# FEDERATED WASSERSTEIN DISTANCE

**Alain Rakotomamonjy**
Criteo AI Lab
Paris, France
alain.rakoto@insa-rouen.fr

**Kimia Nadjahi**
CSAIL, MIT
Boston, MA
knadjahi@mit.edu

**Liva Ralaivola**
Criteo AI Lab
Paris, France
l.ralaivola@criteo.com

## ABSTRACT

We introduce a principled way of computing the Wasserstein distance between two distributions in a federated manner. Namely, we show how to estimate the Wasserstein distance between two samples stored and kept on different devices/clients whilst a central entity/server orchestrates the computations (again, without having access to the samples). To achieve this feat, we take advantage of the geometric properties of the Wasserstein distance – in particular, the triangle inequality – and that of the associated *geodesics*: our algorithm, `FedWaD` (for Federated Wasserstein Distance), iteratively approximates the Wasserstein distance by manipulating and exchanging distributions from the space of geodesics in lieu of the input samples. In addition to establishing the convergence properties of `FedWaD`, we provide empirical results on federated coresets and federate optimal transport dataset distance, that we respectively exploit for building a novel federated model and for boosting performance of popular federated learning algorithms.

## 1 INTRODUCTION

**Context.** Federated Learning (FL) is a form of distributed machine learning (ML) dedicated to train a global model from data stored on local devices/clients, while ensuring these clients never share their data (Kairouz et al., 2021; Wang et al., 2021). FL provides elegant and convenient solutions to concerns in data privacy, computational and storage costs of centralized training, and makes it possible to take advantage of large amounts of data stored on local devices. A typical FL approach to learn a parameterized global model is to alternate between the two following steps: i) update local versions of the global model using local data, and ii) send and aggregate the parameters of the local models on a central server (McMahan et al., 2017) to update the global model.

**Problem.** In some practical situations, the goal is not to learn a prediction model, but rather to compute a certain quantity from the data stored on the clients. For instance, one's goal may be to compute, in a federated way, some prototypes of client's data, that can be leveraged for federated clustering or for classification models (Gribonval et al., 2021; Phillips, 2016; Munteanu et al., 2018; Agarwal et al., 2005). In another learning scenarios where data are scarce, one may want to look for similarity between datasets in order to evaluate dataset heterogeneity over clients and leverage on this information to improve the performance of federated learning algorithms. In this work, we address the problem of computing, in a federated way, the Wasserstein distance between two distributions $\mu$ and $\nu$ when samples from each distribution are stored on local devices. A solution to this problem will be useful in the aforementioned situations, where the Wasserstein distance is used as a similarity measure between two datasets and is the key tool for computing some coresets of the data distribution or cluster prototypes. We provide a solution to this problem which hinges on the geometry of the Wasserstein distance and more specifically, its geodesics. We leverage the property that for any element $\xi^\star$ of the geodesic between two distributions $\mu$ and $\nu$, the following equality holds, $\mathcal{W}_p(\mu, \nu) = \mathcal{W}_p(\mu, \xi^\star) + \mathcal{W}_p(\xi^\star, \nu)$, where $\mathcal{W}_p$ denotes the $p$-Wasserstein distance. This property is especially useful to compute $\mathcal{W}_p(\mu, \nu)$ in a federated manner, leading to a novel theoretically-justified procedure coined `FedWaD`, for **Fed**erated **Wa**sserstein **D**istance.

**Contribution: `FedWaD`.** The principle of `FedWaD` is to iteratively approximate $\xi^\star$ – which, in terms of traditional FL, can be interpreted as the global model. At iteration $k$, our procedure consists in i) computing, on the clients, distributions $\xi_\mu^k$ and $\xi_\nu^k$ from the geodesics between the current

approximation of $\xi^\star$ and the two secluded distributions $\mu$ and $\nu - \xi_\mu^k$ and $\xi_\nu^k$ playing the role of the local versions of the global model, and ii) aggregating them on the global model to update $\xi^\star$.

**Organization of the paper.** Section 2 formalizes the problem we address, and provides the necessary technical background to devise our algorithm `FedWaD`. Section 3 is devoted to the depiction of `FedWaD`, pathways to speed-up its executions, and a theoretical justification that `FedWaD` is guaranteed to converge to the desired quantity. In Section 4, we conduct an empirical analysis of `FedWaD` on different use-cases (Wasserstein coresets and Optimal Transport Dataset distance) which rely on the computation of the Wasserstein distance. We unveil how these problems can be solved in our FL setting and demonstrates the remarkable versatility of our approach. In particular, we expose the impact of federated coresets. By learning a single global model on the server based on the coreset, our method can outperform personalized FL models. In addition, our ability to compute inter-device dataset distances significantly helps amplify performances of popular federated learning algorithms, such as FedAvg, FedRep, and FedPer. We achieve this by clustering clients and harnessing the power of reduced dataset heterogeneity.

## 2 RELATED WORKS AND BACKGROUND

### 2.1 WASSERSTEIN DISTANCE AND GEODESICS

Throughout, we denote by $\mathscr{P}(X)$ the set of probability measures in $X$. Let $p \geq 1$ and define $\mathscr{P}_p(X)$ the subset of measures in $\mathscr{P}(X)$ with finite $p$-moment, *i.e.*, $\mathscr{P}_p(X) \doteq \{\eta \in \mathscr{P}(X) : M_p(\eta) < \infty\}$, where $M_p(\eta) \doteq \int_X d_X^p(x, 0) d\eta(x)$ and $d_X$ is a metric on $X$ often referred to as the *ground cost*. For $\mu \in \mathscr{P}_p(X)$ and $\nu \in \mathscr{P}_p(Y)$, $\Pi(\mu, \nu) \subset \mathscr{P}(X \times Y)$ is the collection of probability measures or *couplings* on $X \times Y$ defined as

$$\Pi(\mu, \nu) \doteq \{\pi \in \mathscr{P}(X \times Y) : \forall A \subset X, B \subset Y, \pi(A \times Y) = \mu(A) \text{ and } \pi(X \times B) = \nu(B)\}.$$

The $p$-Wasserstein distance $\mathcal{W}_p(\mu, \nu)$ between the measures $\mu$ and $\nu$ —assumed to be defined over the same ground space, i.e. $X = Y$— is defined as

$$\mathcal{W}_p(\mu, \nu) \doteq \left( \inf_{\pi \in \Pi(\mu, \nu)} \int_{X \times X} d_X^p(x, x') d\pi(x, x') \right)^{1/p}. \tag{1}$$

It is proven that the infimum in (1) is attained (Peyré et al., 2019) and any probability $\pi$ which realizes the minimum is an *optimal transport plan*. In the discrete case, we denote the two marginal measures as $\mu = \sum_{i=1}^n a_i \delta_{x_i}$ and $\nu = \sum_{i=1}^m b_i \delta_{x'_i}$, with $a_i, b_i \geq 0$ and $\sum_{i=1}^n a_i = \sum_{i=1}^m b_i = 1$. The *Kantorovitch relaxation* of (1) seeks for a transportation coupling $\mathbf{P}$ that solves the problem

$$\mathcal{W}_p(\mu, \nu) \doteq \left( \min_{\mathbf{P} \in \Pi(\mathbf{a}, \mathbf{b})} \langle \mathbf{C}, \mathbf{P} \rangle \right)^{1/p} \tag{2}$$

where $\mathbf{C} \doteq (d_X^p(x_i, x'_j)) \in \mathbb{R}^{n \times m}$ is the matrix of all pairwise costs, and $\Pi(\mathbf{a}, \mathbf{b}) \doteq \{\mathbf{P} \in \mathbb{R}_+^{n \times m} | \mathbf{P}\mathbf{1} = \mathbf{a}, \mathbf{P}^\top \mathbf{1} = \mathbf{b}\}$ is the *transportation polytope* (i.e. the set of all transportation plans) between the distributions $\mathbf{a}$ and $\mathbf{b}$.

**Property 1** (Peyré et al. (2019)). *For any $p \geq 1$, $\mathcal{W}_p$ is a metric on $\mathscr{P}_p(X)$. As such it satisfies the triangle inequality:*

$$\forall \mu, \nu, \xi \in \mathscr{P}_p(X), \quad \mathcal{W}_p(\mu, \nu) \leq \mathcal{W}_p(\mu, \xi) + \mathcal{W}_p(\xi, \nu) \tag{3}$$

It might be convenient to consider *geodesics* as structuring tools of metric spaces.

**Definition 1** (Geodesics, Ambrosio et al. (2005)). *Let $(\mathcal{X}, d)$ be a metric space. A* constant speed geodesic $x : [0, 1] \to \mathcal{X}$ *between $x_0, x_1 \in \mathcal{X}$ is a continuous curve such that $\forall s, t \in [0, 1]$, $d(x(s), x(t)) = |s - t| \cdot d(x_0, x_1)$.*

**Property 2** (Interpolating point, Ambrosio et al. (2005)). *Any point $x_t$ from a constant speed geodesic $(x(t))_{t \in [0,1]}$ is an* interpolating point *and verifies, $d(x_0, x_1) = d(x_0, x_t) + d(x_t, x_1)$, i.e. the triangle inequality becomes an equality.*

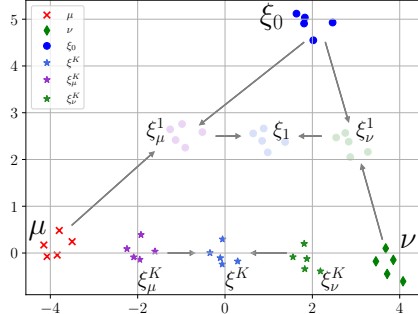

- At first iteration the current estimation $\xi_0$ of $\xi^\star$ is sent to each client in order to compute two interpolating measures $\xi_\mu^1$ and $\xi_\nu^1$, which are sent back to the server.

- Server then computes an interpolating measure between $\xi_\mu^1$ and $\xi_\nu^1$ to obtain the next iterate of the geodesic element $\xi^1$

- The process is repeated until convergence to obtain $\xi^K$ and we define $\mathcal{W}_p(\mu, \nu) = \mathcal{W}_p(\mu, \xi^K) + \mathcal{W}_p(\nu, \xi^K)$.

Figure 1: The Wasserstein distance between $\mu$ and $\nu$ which are on their respective clients can be computed as $\mathcal{W}_p(\mu, \nu) = \mathcal{W}_p(\mu, \xi^\star) + \mathcal{W}_p(\nu, \xi^\star)$ where $\xi^\star$ is an element on the geodesic between $\mu$ and $\nu$. `FedWaD` seeks at estimating $\xi^\star$ with $\xi^K$ using an iterative algorithm and plugs in this estimation to obtain $\mathcal{W}_p(\mu, \nu)$. Iterates of $\xi_i$ are computed on the server and sent to clients in order to compute measures $\xi_\mu^i$ and $\xi_\nu^i$ that are on geodesics of $\mu$ and $\xi_i$ and $\nu$ and $\xi_i$ respectively.

These definitions and properties carry over to the case of the Wasserstein distance:

**Definition 2** (Wasserstein Geodesics, Interpolating measure, Ambrosio et al. (2005); Kolouri et al. (2017)). *Let $\mu_0$, $\mu_1 \in \mathscr{P}_p(X)$ with $X \subseteq \mathbb{R}^d$ compact, convex and equipped with $\mathcal{W}_p$. Let $\gamma \in \Pi(\mu_0, \mu_1)$ be an optimal transport plan. For $t \in [0, 1]$, let $\mu_t \doteq (\pi_t)_{\#}\gamma$ where $\pi_t(x, y) \doteq (1 - t)x + ty$, i.e. $\mu_t$ is the push-forward measure of $\gamma$ under the map $\pi_t$. Then, the curve $\bar{\mu} \doteq (\mu_t)_{t \in [0,1]}$ is a constant speed geodesic between $\mu_0$ and $\mu_1$; we call it a* Wasserstein geodesics *between $\mu_0$ and $\mu_1$.*

*Any point $\mu_t$ of the geodesics is an* interpolating measure *between $\mu_0$ and $\mu_1$ and, as expected:*

$$\mathcal{W}_p(\mu_0, \mu_1) = \mathcal{W}_p(\mu_0, \mu_t) + \mathcal{W}_p(\mu_t, \mu_1). \tag{4}$$

In the discrete case, and for a fixed $t$, one can obtain such interpolating measure $\mu_t$ given the optimal transport map $\mathbf{P}^\star$ solution of Equation (2) as follows (Peyré et al., 2019, Remark 7.1):

$$\mu_t = \sum_{i,j}^{n,m} \mathbf{P}_{i,j}^\star \delta_{(1-t)x_i + tx_j'} \tag{5}$$

where $\mathbf{P}_{i,j}^\star$ is the $(i, j)$-th entry of $\mathbf{P}^\star$; as an interpolating measure, $\mu_t$ obviously complies with (4).

## 2.2 PROBLEM STATEMENT

Our goal is to compute the Wasserstein distance between two data distributions $\mu$ and $\nu$ on a global server with the constraint that $\mu$ and $\nu$ are distributed on two different clients which do not share any data samples to the server. From a mathematical point of view, our objective is to estimate an element $\xi^\star$ on the geodesic of $\mu$ and $\nu$ without having access to them by leveraging two other elements $\xi_\mu$ and $\xi_\nu$ on the geodesics of $\mu$ and $\xi^\star$ and $\nu$ and $\xi^\star$ respectively.

## 2.3 RELATED WORKS

Our work touches the specific question of learning/approximating a distance between distributions whose samples are secluded on isolated clients. As far as we are aware of, this is a problem that has never been investigated before and there are only few works that we see closely connected to ours. Some problems have addressed the objective of retrieving nearest neighbours of one vector in a federated manner. For instance, Liu et al. (2021) consider to exchange encrypted versions of the dataset on client to the central server and Schoppmann et al. (2018) consider the exchange of differentially private statistics about the client dataset. Zhang et al. (2023) propose a federated approximate $k$-nearest approach based on a specific spatial data federation. Compared to these works that compute distances in a federated manner, we address the case of distances on distributions without any specific encryption of the data and we exploit the properties of the Wasserstein distances and its geodesics, which have been overlooked in the mentioned works. If these properties have been

relied upon as a key tool in some computer vision applications (Bauer et al., 2015; Maas et al., 2017) and trajectory inference (Huguet et al., 2022), they have not been employed as a privacy-preserving tool.

## 3 Computing the Federated Wasserstein distance

In this section, we develop a methodology to compute, on a global server, the Wasserstein distance between two distributions $\mu$ and $\nu$, stored on two different clients which do not share this information to the server. Our approach leverages the topology induced by the Wasserstein distance in the space of probability measures, and more precisely, the geodesics.

**Outline of our methodology.** A key property is that $\mathcal{W}_p$ is a metric, thus satisfies the triangle inequality: for any $\mu, \nu, \xi \in \mathscr{P}_p(X)$,

$$\mathcal{W}_p(\mu, \nu) \leq \mathcal{W}_p(\mu, \xi) + \mathcal{W}_p(\xi, \nu), \tag{6}$$

with equality if and only if $\xi = \xi^\star$, where $\xi^\star$ is an interpolating measure. Consequently, one can compute $\mathcal{W}_p(\mu, \nu)$ by computing $\mathcal{W}_p(\mu, \xi^\star)$ and $\mathcal{W}_p(\xi^\star, \nu)$ and adding these two terms. This result is useful in the federated setting and inspires our methodology, as described hereafter. The global server computes $\xi^\star$ and communicate $\xi^\star$ to the two clients. The clients respectively compute $\mathcal{W}_p(\mu, \xi^\star)$ and $\mathcal{W}_p(\xi^\star, \nu)$, then send these to the global server. Finally, the global server adds the two received terms to return $\mathcal{W}_p(\mu, \nu)$.

The main bottleneck of this procedure is that the global server needs to compute $\xi^\star$ (which by definition, depends on $\mu, \nu$) while not having access to $\mu, \nu$ (which are stored on two clients). We then propose a simple workaround to overcome this challenge, based on an additional application of the triangle inequality: for any $\xi \in \mathscr{P}_p(X)$,

$$\mathcal{W}_p(\mu, \nu) \leq \mathcal{W}_p(\mu, \xi) + \mathcal{W}_p(\xi, \nu) = \mathcal{W}_p(\mu, \xi_\mu) + \mathcal{W}_p(\xi_\mu, \xi) + \mathcal{W}_p(\xi, \xi_\nu) + \mathcal{W}_p(\xi_\nu, \nu), \tag{7}$$

where $\xi_\mu$ and $\xi_\nu$ are interpolating measures respectively between $\mu$ and $\xi$ and $\xi$ and $\nu$. Hence, computing $\xi^\star$ can be done through intermediate measures $\xi_\mu$ and $\xi_\nu$, to ensure that $\mu, \nu$ stay on their respective clients. To this end, we develop an optimization procedure which essentially consists in iteratively estimating an interpolating measure $\xi^{(k)}$ between $\mu$ and $\nu$ on the server, by using $\xi_\mu^{(k)}$ and $\xi_\nu^{(k)}$ which were computed and communicated by the clients. More precisely, the objective is to minimize (7) over $\xi$ as follows: at iteration $k$, the clients receive current iterate $\xi^{(k-1)}$ and compute $\xi_\mu^{(k)}$ and $\xi_\nu^{(k)}$ (as interpolating measures between $\mu$ and $\xi^{(k-1)}$, and between $\xi^{(k-1)}$ and $\nu$ respectively). By the triangle inequality,

$$\mathcal{W}_p(\mu, \nu) \leq \mathcal{W}_p(\mu, \xi_\mu^{(k)}) + \mathcal{W}_p(\xi_\mu^{(k)}, \xi^{(k-1)}) + \mathcal{W}_p(\xi^{(k-1)}, \xi_\nu^{(k)}) + \mathcal{W}_p(\xi_\nu^{(k)}, \nu), \tag{8}$$

therefore, the clients then send $\xi_\mu^{(k)}$ and $\xi_\nu^{(k)}$ to the server, which in turn, computes the next iterate $\xi^{(k)}$ by minimizing the left-hand side term of (8), *i.e.*,

$$\xi^{(k)} \in \arg\min_{\xi} \mathcal{W}_p(\xi_\mu^{(k)}, \xi) + \mathcal{W}_p(\xi, \xi_\nu^{(k)}). \tag{9}$$

Our methodology is illustrated in Figure 1 and summarized in Algorithm 1. It can be applied to continuous measures as long as an interpolating measure between two distributions can be computed in closed form. Regarding communication cost, at each iteration, the communication cost involves the transfer between the server and the clients of four interpolating measures: $\xi^{(k-1)}$ (twice), $\xi_\mu^{(k)}$, $\xi_\nu^{(k)}$. Hence, if the support size of $\xi^{(k-1)}$ is $S$, the communication cost is in $\mathcal{O}(4SKd)$, with $d$ the data dimension and $K$ the number of iterations.

**Reducing the computational complexity.** In terms of computational complexity, we need to compute three OT plans per iteration which single cost, based on the network simplex is $O((n + m)nm\log(n + m))$. More importantly, consider that $\mu$ and $\nu$ are discrete measures, then, any interpolating measure between $\mu$ and $\nu$ is supported on at most on $n + m + 1$ points. Hence, even if the size of the support of $\xi^{(0)}$ is small, but $n$ is large, the support of the next interpolating measures may get larger and larger, and this can yield an important computational overhead when computing $\mathcal{W}_p(\mu, \xi^{(k)})$ and $\mathcal{W}_p(\xi^{(k)}, \nu)$.

To reduce this complexity, we resort to approximations of the interpolating measures which goal is to fix the support size of the interpolating measures to a small number $S$. The solution we consider is to approximate the McCann's interpolation equation which formalizes geodesics $\xi_t$ given an optimal transport map between two distributions,say, $\xi$ and $\xi'$, based on the equation $\xi_t = ((1-t)Id + tT)_\# \xi$ Peyré et al. (2019). Using barycentric mapping approximation of the map $T$ (Courty et al., 2018), we propose to approximate the interpolating measures $\xi_t$ as

$$\xi_t = \frac{1}{n}\sum_{i=1}^{n} \delta_{(1-t)x_i + tn(\mathbf{P}^\star \mathbf{X}')_i} \qquad (10)$$

where $\mathbf{P}^\star$ is the optimal transportation plan between $\xi$ and $\xi'$, $x_i$ and $x'_j$ are the samples from these distributions and $\mathbf{X}'$ is the matrix of samples from $\xi'$. Note that by choosing the appro-

---

**Algorithm 1** `FedWaD`

**Input:** $\mu$ and $\nu$, initialisation of $\xi^{(0)}$, function *Interp-Meas* that computes an interpolating measure between two measures using Equation (5) or Equation (10) for any $0 < t < 1$.
1: **for** $k = 1$ **to** $K$ **do**
2:     *// Send $\xi^{(k-1)}$ to clients*
3:     *// Compute on clients with optional return of distances*
4:     $\xi_\mu^{(k)}, [\mathcal{W}_p(\mu, \xi^{(k)})] \leftarrow \texttt{InterpMeas}(\mu, \xi^{(k-1)})$
5:     $\xi_\nu^{(k)}, [\mathcal{W}_p(\xi^{(k)}, \nu)] \leftarrow \texttt{InterpMeas}(\nu, \xi^{(k-1)})$
6:     *// Send $\xi_\mu^{(k)}$ and $\xi_\nu^{(k)}$ to server*
7:     $\xi^{(k)} \leftarrow \texttt{InterpMeas}(\xi_\mu^{(k)}, \xi_\nu^{(k)})$
8: **end for**
9: *// Send $\mathcal{W}_p(\mu, \xi^{(K)})$, $\mathcal{W}_p(\xi^{(K)}, \nu)$ to server*
10: $\mathcal{W}_p(\mu, \nu) = \mathcal{W}_p(\mu, \xi^{(K)}) + \mathcal{W}_p(\xi^{(K)}, \nu)$
**Output:** return $d_{\mu,\nu}$ on server

---

priate formulation of the equation, the support size of this interpolating measure can be chosen as the one of $\xi$ or $\xi'$. In practice, we always opt for the choice that leads to the smallest support of the interpolating measure. Hence, if the support size of $\xi^{(0)}$ is $S$, we have the guarantee that the support of $\xi^{(k)}$ is $S$ for all $k$. Then, for computing $\mathcal{W}_p(\mu, \xi^{(k)})$ using approximated interpolating measures, it costs $O(3*(Sn^2 + S^2n)log(n+S))$ at each iteration and if $S$ and the number of iterations $K$ are small enough, the approach we propose is even competitive compared to exact OT. Our experiments reported later that for larger number of samples ($\geq 5000$), our approach is as fast as exact optimal transport and less prone to numerical errors.

**Mitigating privacy issues**. As for many FL algorithms, we do not provide or have a formal guarantee of privacy. However, we have components of the algorithm that helps mitigate risks of privacy leak. First, the interpolating measures can be computed for a randomized value of $t$; second, distances are not communicated to the server until the last iteration, and finally the use of the approximated interpolating measures in Equation (10) helps in obfuscation since interpolating measure supports depend on the transport plan which is not reveal to the server. If a formal differential privacy guarantee is required, one need to incorporate an (adapted) differentially private version of the Wasserstein distance (Lê Tien et al., 2019; Goldfeld & Greenewald, 2020).

**Theoretical guarantees.** We discuss in this section some theoretical properties of the components of `FedWaD`. At first, we show that the approximated interpolating measure is tight in the sense that there exists some situations where the resulting approximation is exact.

**Theorem 1.** *Consider two discrete distributions $\mu$ and $\nu$ with the same number of samples $n$ and uniform weights, then for any $t$, the approximated interpolating measure, between $\mu$ and $\nu$ given by Equation* (10) *is equal to the exact one Equation* (5).

Proof is given in Appendix A. In practice, this property does not have much impact, but it ensures us about the soundness of the approach. In the next theorem, we prove that Algorithm 1 is theoretically justified, in the sense that its output converges to $\mathcal{W}_p(\mu, \nu)$.

**Theorem 2.** *Let $\mu$ and $\nu$ be two measures in $\mathscr{P}_p(X)$, $\xi_\mu^{(k)}$, $\xi_\nu^{(k)}$ and $\xi^{(k)}$ be the interpolating measures computed at iteration $k$ as defined in Algorithm 1. Denote as*

$$A^{(k)} = \mathcal{W}_p(\mu, \xi_\mu^{(k)}) + \mathcal{W}_p(\xi_\mu^{(k)}, \xi^{(k)}) + \mathcal{W}_p(\xi^{(k)}, \xi_\nu^{(k)}) + \mathcal{W}_p(\xi_\nu^{(k)}, \nu)$$

*Then the sequence $(A^{(k)})_k$ is non-increasing and converges to $\mathcal{W}_p(\mu, \nu)$.*

We provide hereafter a sketch of the proof, and refer to Appendix B for full details. First, we show that the sequence $(A^{(k)})_k$ is non-increasing, as we iteratively update $\xi_\mu^{(k+1)}$, $\xi_\nu^{(k+1)}$ and $\xi^{(k+1)}$ based on geodesics (a minimizer of the triangle inequality). Then, we show that the sequence $(A^{(k)})_k$ is

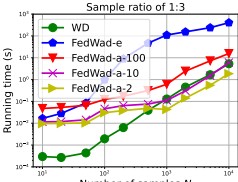 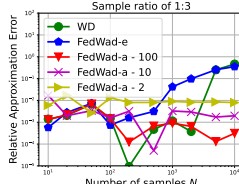 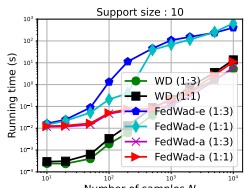 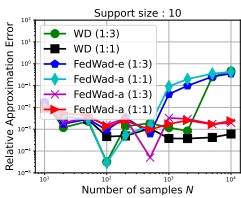

Figure 2: Analysis of the different Wasserstein distance computation methods (most-left panels) for varying support size of the approximated `FedWaD` and (most-right panels) for varying sample ratio in the two distributions and fixed support size. For each couple of panels, for increasing number of samples, we report the running time and the relative error of the Wasserstein distance (WD), our exact FedWaD (FedWad-e) and our approximate FedWaD (FedWad-a) with a support size of 2, 10 and 100. For the most-right panels, we have set the support size of the interpolating measure to 10. For a sample ratio (1:3), the first distribution has a number of samples $N$ and the second ones $N/3$.

bounded below by $\mathcal{W}_p(\mu, \nu)$. We conclude the proof by proving that the sequence $(A^{(k)})_k$ converges to $\mathcal{W}_p(\mu, \nu)$.

In the next theorem, we show that when $\mu$ and $\nu$ are Gaussians then we can recover some nicer properties of our algorithm and provide a convergence rate (proof in Appendix C).

**Theorem 3.** *Assume that $\mu$, $\nu$ and $\xi^{(0)}$ are three Gaussian distributions with the same covariance matrix $\Sigma$ ie $\mu \sim \mathcal{N}(\mathbf{m}_\mu, \Sigma)$, $\nu \sim \mathcal{N}(\mathbf{m}_\nu, \Sigma)$ and $\xi^{(0)} \sim \mathcal{N}(\mathbf{m}_{\xi^{(0)}}, \Sigma)$. Further assume that we are not in the trivial case where $\mathbf{m}_\mu$, $\mathbf{m}_\nu$, and $\mathbf{m}_{\xi^{(0)}}$ are aligned. Applying our Algorithm 1 with $t = 0.5$ and the squared Euclidean cost, we have the following properties:*

1. *all interpolating measures $\xi_\mu^{(k)}, \xi_\nu^{(k)}, \xi^{(k)}$ are Gaussian distributions with the same covariance matrix $\Sigma$,*

2. *for any $k \geq 1$, $\mathcal{W}_2(\mu, \nu) = \|\mathbf{m}_\mu - \mathbf{m}_\nu\|_2 = 2\|\mathbf{m}_{\xi_\mu^{(k)}} - \mathbf{m}_{\xi_\nu^{(k)}}\|_2 = 2\mathcal{W}_2(\xi_\mu^{(k)}, \xi_\nu^{(k)})$*

3. *$\mathcal{W}_2(\xi^{(k)}, \xi^\star) = \frac{1}{2}\mathcal{W}_2(\xi^{(k-1)}, \xi^\star)$*

4. *$\mathcal{W}_2(\mu, \xi^{(k)}) + \mathcal{W}_2(\xi^{(k)}, \nu) - \mathcal{W}_2(\mu, \nu) \leq \frac{1}{2^{k-1}}\mathcal{W}_2(\xi^{(0)}, \xi^{(\star)})$*

Interestingly, this theorem also says that in this specific case, only one iteration is needed to recover $\mathcal{W}_2(\mu, \nu)$

## 4 EXPERIMENTS

This section presents numerical applications, where `FedWaD` can successfully be used and show how it can boost performances of federated learning algorithms. The code for reproducing part of the results is available at `https://github.com/arakotom/fedwad` and is built on top of the Python Optimal Transport library (Flamary et al., 2021). Full details are provided in Appendix D.

**Toy analysis.** We illustrate the evolution of interpolating measures using `FedWaD` for calculating the Wasserstein distance between two Gaussian distributions. We sample 200 points from two 2D Gaussian distributions with different means and the same covariance matrix. We compute the interpolating measure at $t = 0.5$ using both the analytical formula (5) and the approximation (10). Figure 3 (left panel) shows how the interpolating measure evolves across iterations. We also observe, in Figure 3 (right panel), that the error on the true Wasserstein distance for the approximated interpolating measure reaches $10^{-3}$, while for the exact interpolating

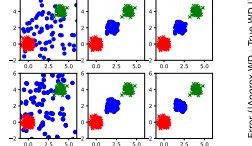 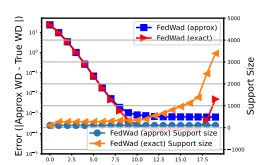

Figure 3: (left) Evolution of the interpolating measure $\xi^{(k)}$ - in blue - (right) the estimated Wasserstein distance between two Gaussian distributions $\mu$ and $\nu$.

measure, it drops to a minimum of $10^{-4}$ before increasing. This discrepancy occurs as the support size of the interpolating measure expands across iterations leading to numerical errors when computing the optimal transport plan between $\xi^{(k)}$ and $\xi_\mu^{(k)}$ or $\xi_\nu^{(k)}$. Hence, using the approximation Equation (10) is a more robust alternative to exact computation Equation (5).

We also examine computational complexity and approximation errors for both methods as we increase sample sizes in the distributions, as displayed in Figure 2. Key findings include: The approximated interpolating measure significantly improves computational efficiency, being at least 10 times faster with sample size exceeding 100, especially with smaller support sizes. It also achieves a similar relative approximation error as `FedWaD` using the exact interpolating measure and true non-federated Wasserstein distance. Importantly, it demonstrates greater robustness with larger sample sizes compared to true Wasserstein distance for such a small dimensional problem.

**Wasserstein coreset and application to federated learning.** In many ML applications, summarizing data into fewer representative samples is routinely done to deal with large datasets. The notion of *coreset* has been relevant to extract such samples and admit several formulations (Phillips, 2016; Munteanu et al., 2018). In this experiment, we show that Wasserstein coresets (Claici et al., 2018) can be computed in a federated way via `FedWaD`. Formally, given a dataset described by the distribution $\mu$, the Wasserstein coreset aims at finding the empirical distribution that minimizes $\min_{x'_1, \cdots, x'_K} \mathcal{W}_p \left( \frac{1}{K} \sum_{i=1}^K \delta_{x'_i}, \mu \right)$. We solve this problem in the following federated setting: we assume that either the samples drawn from $\mu$ are stored on an unique client or distributed across different clients, and the objective is to learn the coreset samples $\{x'_i\}$ on the server. In our setting, we can compute the federated Wasserstein distances between the current coreset and some subsamples of all active client datasets, then update the coreset given the aggregated gradients of these distances with respect to the coreset support. We sampled 20000 examples randomly from the MNIST dataset, and dispatched them at

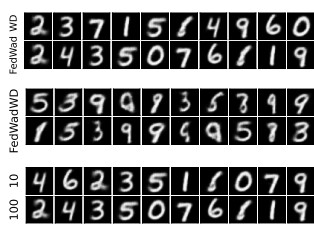

Figure 4: Examples of the 10 coreset we obtained, with for each panel *(top-row)* the exact Wasserstein and *(bottow-row)* `FedWaD` for the MNIST dataset. Different panels correspond to different number of classes $K$ on each client: *(top)* $K = 8$, *(middle)* $K = 2$, *(bottom)* support of the interpolating measure varying from 10 to 100.

random on 100 clients. We compare the results we obtained with `FedWaD` with those obtained with exact non-federated Wasserstein distance The results are shown in Figure 4. We can note that when classes are almost equally spread across clients (with $K = 8$ different classes per client), `FedWaD` is able to capture the 10 modes of the dataset. However, as the diversity in classes between clients increases, `FedWaD` has more difficulty to capture all the modes of the dataset. Nonetheless, we also observe that the exact Wasserstein distance is not able to recover those modes either. We can thus conjecture that this failure is likely due to the coreset approach itself, rather than to the approximated distance returned by `FedWaD`. We also note that the support size of the interpolating measure has less impact on the coreset. We believe this is a very interesting result, as it shows that `FedWaD` can provide useful gradient to the problem even with a poorer estimation of the distance.

**Federated coreset classification model** Those federated coresets can also be used for classification tasks. As such, we have learned coresets for each client, and used all the coresets from all clients as the examples for a one-nearest neighbor global classifier shared to all clients. Note that since a coreset computation is an unsupervised task, we have assigned to each element of a coreset the label of the closest element in the client dataset. For this task, we have used the MNIST dataset which has been autoencoded in order to reduce its dimensionality. Half of the training samples have been used for learning the autoencoder and the other half for the classification task. Those samples and the test samples of dataset have been distributed across clients while ensuring that each client has samples from only 2 classes. We have then computed the accuracy of this federated classifier for varying number of clients and number of coresets and compared its performance to the ones of *FedRep* (Collins et al., 2021) and *FedPer* (Arivazhagan et al., 2019). Results are reported in Figure 5. We can see that our simple approach is highly competitive with these personalized FL approaches, and even outperforms them when the number of users becomes large.

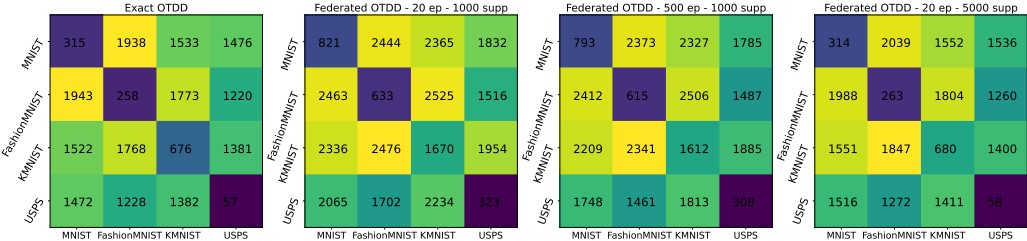

Figure 6: Comparison of the matrix of distances between digits datasets computed by `FedWaD` and the true OTDD distance between the same datasets. *(left)* OTDD, *(middle-left)* `FedWaD` with 20 epochs and 1000 support points, *(middle-right)* `FedWaD` with 500 epochs and 1000 support points, *(right)* `FedWaD` with 20 epochs and 5000 support points

**Geometric dataset distances via federated Wasserstein distance.** Our goal is to improve on the seminal algorithm of Alvarez-Melis & Fusi (2020) that seeks at computing distance between two datasets $\mathcal{D}$ and $\mathcal{D}'$ using optimal transport. We want to make it federated. This extension will pave the way to better federated learning algorithms for transfer learning and domain adaptation or can simply be used for boosting federated learning algorithms, as we illustrate next. Alvarez-Melis & Fusi (2020) considers a Wasserstein distance with a ground metric that mixes distances between features and tractable distance between class-conditional distributions. For our extension, we will use the same ground metric, but we will compute the Wasserstein distance using `FedWaD`. Details are provided in Appendix D.5.

We replicated the experiments of Alvarez-Melis & Fusi (2020) on the dataset selection for transfer learning: given a source dataset, the goal is to find a target one which is the most similar to the source. We considered four real

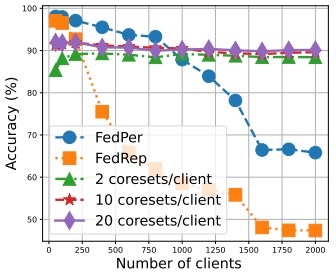

Figure 5: Nearest neighbor classifier based on the coresets learnt from each client for varying number of clients and number of coresets per clients We have compared to the performance of two personalized FL algorithms.

datasets, namely `MNIST`, `KMNIST`, `USPS` and `FashionMNIST` and we have computed all the pairwise distance between 5000 randomly selected examples from each dataset using the original `OTDD` of Alvarez-Melis & Fusi (2020) and our `FedWaD` approach. For `FedWaD`, we chose the support size of the interpolating measure to be 1000 and 5000 and the number of epochs to 20 and 500. Results, averaged over 5 random draw of the samples, are depicted in Figure 6. We can see that the distance matrices produced by `FedWaD` are semantically similar to the ones for OTDD distance, which means that order relations are well-preserved for most pairwise distances (except only for two pairs of datasets in the `USPS` row). More importantly, running more epochs leads to slightly better approximation of the OTDD distance, but the exact order relations are already uncovered using only 20 epochs in `FedWaD`. Detailed ablation studies on these parameters are provided in Appendix D.6.

**Boosting FL methods** One of the challenges in FL is the heterogeneity of the data distribution among clients. This heterogeneity is usually due to shift in class-conditional distributions or to a label shift (some classes being absent on a client). As such, we propose to investigate a simple approach that allows to address dataset heterogeneity (in terms of distributions) among clients, by leveraging on our ability to compute distance between datasets in a federated way.

Our proposal involves computing pairwise dataset distances between clients, clustering them based on their (di)-similarities using a spectral clustering algorithm (Von Luxburg, 2007), and using this clustering knowledge to enhance existing federated learning algorithms. In our approach, we run the FL algorithm for each of the $K$ clusters of clients instead of all clients to avoid information exchange between clients with diverse datasets. For example, for FedAvg, this means learning a

Table 1: MNIST/CIFAR10 Average performances over 5 trials of three FL algorithms, FedAvg, FedRep and FedPer. For each algorithm we compare the vanilla performance with the ones obtained after clustering the clients using the `FedWaD` OTDD distance and three different setting of the spectral clustering algorithm (details in Appendix) and for a support size of 10. The number of clients varies from 20 to 100. Bolded number indicate the best performing approach (and clustering parameters).

| | Strong structure | | | | No structure | | | |
|---|---|---|---|---|---|---|---|---|
| | | Clustering | | | | | Clustering | | |
| | Vanilla | Affinity | Sparse G. (3) | Sparse G. (5) | Vanilla | Affinity | Sparse G. (3) | Sparse G. (5) |
| MNIST | | | | | | | | |
| **FedAvg** | | | | | | | | |
| 20 | $26.3 \pm 3.8$ | $\mathbf{99.5 \pm 0.0}$ | $99.5 \pm 0.0$ | $91.5 \pm 10.3$ | $25.1 \pm 6.6$ | $\mathbf{71.3 \pm 7.3}$ | $59.5 \pm 3.0$ | $57.0 \pm 4.4$ |
| 40 | $39.1 \pm 9.0$ | $\mathbf{99.2 \pm 0.1}$ | $91.1 \pm 6.5$ | $94.5 \pm 9.4$ | $42.5 \pm 10.5$ | $\mathbf{70.8 \pm 13.5}$ | $60.0 \pm 3.7$ | $58.1 \pm 6.3$ |
| 100 | $39.2 \pm 7.7$ | $\mathbf{98.9 \pm 0.0}$ | $95.9 \pm 4.6$ | $98.4 \pm 0.8$ | $52.6 \pm 3.9$ | $64.4 \pm 9.6$ | $\mathbf{76.3 \pm 5.4}$ | $67.9 \pm 6.0$ |
| **FedRep** | | | | | | | | |
| 20 | $81.1 \pm 8.1$ | $\mathbf{99.1 \pm 0.0}$ | $99.1 \pm 0.0$ | $98.2 \pm 1.3$ | $75.6 \pm 9.3$ | $\mathbf{87.5 \pm 4.5}$ | $81.4 \pm 8.6$ | $85.3 \pm 7.3$ |
| 40 | $88.8 \pm 10.4$ | $\mathbf{98.9 \pm 0.1}$ | $93.3 \pm 7.1$ | $96.7 \pm 4.5$ | $78.0 \pm 6.3$ | $\mathbf{88.0 \pm 4.3}$ | $78.9 \pm 7.9$ | $76.7 \pm 5.6$ |
| 100 | $93.0 \pm 3.9$ | $\mathbf{98.6 \pm 0.1}$ | $98.4 \pm 0.1$ | $98.5 \pm 0.1$ | $86.0 \pm 4.8$ | $\mathbf{91.6 \pm 3.1}$ | $89.1 \pm 5.0$ | $86.3 \pm 4.9$ |
| **FedPer** | | | | | | | | |
| 20 | $94.3 \pm 4.3$ | $\mathbf{99.5 \pm 0.0}$ | $99.5 \pm 0.0$ | $99.3 \pm 0.3$ | $90.5 \pm 2.4$ | $92.7 \pm 1.5$ | $93.0 \pm 4.3$ | $\mathbf{93.8 \pm 2.9}$ |
| 40 | $94.7 \pm 7.6$ | $\mathbf{99.2 \pm 0.1}$ | $99.1 \pm 0.2$ | $97.9 \pm 2.7$ | $\mathbf{92.3 \pm 1.3}$ | $90.2 \pm 4.7$ | $87.7 \pm 4.1$ | $89.2 \pm 2.3$ |
| 100 | $98.1 \pm 0.1$ | $\mathbf{98.9 \pm 0.0}$ | $98.8 \pm 0.2$ | $\mathbf{98.9 \pm 0.0}$ | $96.6 \pm 0.9$ | $96.6 \pm 1.6$ | $92.1 \pm 3.3$ | $90.2 \pm 4.9$ |
| Average Uplift | - | $\mathbf{26.4 \pm 27.5}$ | $24.4 \pm 26.5$ | $24.4 \pm 25.6$ | - | $\mathbf{12.7 \pm 14.6}$ | $8.7 \pm 12.7$ | $7.2 \pm 11.4$ |
| CIFAR10 | | | | | | | | |
| **FedAvg** | | | | | | | | |
| 20 | $22.0 \pm 2.6$ | $\mathbf{75.1 \pm 6.2}$ | $42.6 \pm 4.5$ | $52.2 \pm 8.8$ | $23.5 \pm 6.9$ | $\mathbf{71.4 \pm 9.7}$ | $42.5 \pm 4.7$ | $49.7 \pm 4.7$ |
| 40 | $26.1 \pm 7.1$ | $\mathbf{65.9 \pm 7.1}$ | $36.7 \pm 18.3$ | $48.8 \pm 8.3$ | $26.6 \pm 5.1$ | $\mathbf{73.4 \pm 15.9}$ | $36.3 \pm 4.5$ | $32.3 \pm 11.6$ |
| 100 | $26.4 \pm 4.3$ | $\mathbf{68.0 \pm 5.1}$ | $37.4 \pm 11.4$ | $39.8 \pm 8.0$ | $27.5 \pm 2.0$ | $\mathbf{54.6 \pm 10.1}$ | $27.6 \pm 4.1$ | $29.0 \pm 3.8$ |
| **FedRep** | | | | | | | | |
| 20 | $81.8 \pm 1.8$ | $\mathbf{88.1 \pm 2.0}$ | $84.4 \pm 0.5$ | $85.3 \pm 0.5$ | $85.3 \pm 2.0$ | $\mathbf{90.7 \pm 2.5}$ | $87.9 \pm 2.0$ | $88.1 \pm 1.4$ |
| 40 | $80.3 \pm 0.8$ | $\mathbf{83.7 \pm 2.0}$ | $81.0 \pm 2.1$ | $81.6 \pm 1.7$ | $84.1 \pm 0.8$ | $\mathbf{93.6 \pm 2.9}$ | $84.8 \pm 1.7$ | $84.3 \pm 0.5$ |
| 100 | $75.0 \pm 0.9$ | $\mathbf{79.4 \pm 2.3}$ | $75.2 \pm 2.4$ | $75.4 \pm 1.5$ | $77.9 \pm 1.4$ | $\mathbf{91.4 \pm 2.0}$ | $77.8 \pm 1.7$ | $79.0 \pm 1.1$ |
| **FedPer** | | | | | | | | |
| 20 | $85.4 \pm 2.3$ | $\mathbf{91.0 \pm 1.9}$ | $87.2 \pm 0.5$ | $87.8 \pm 0.9$ | $88.7 \pm 1.7$ | $\mathbf{92.3 \pm 1.8}$ | $89.8 \pm 2.0$ | $90.1 \pm 1.5$ |
| 40 | $85.9 \pm 0.8$ | $\mathbf{87.2 \pm 2.2}$ | $82.7 \pm 2.5$ | $84.3 \pm 1.9$ | $88.1 \pm 0.7$ | $\mathbf{94.8 \pm 2.6}$ | $86.0 \pm 2.3$ | $84.9 \pm 3.3$ |
| 100 | $82.2 \pm 0.4$ | $\mathbf{85.1 \pm 1.8}$ | $80.3 \pm 2.0$ | $80.9 \pm 1.7$ | $85.1 \pm 0.6$ | $\mathbf{94.0 \pm 1.4}$ | $82.0 \pm 2.4$ | $83.0 \pm 1.1$ |
| Average Uplift | - | $\mathbf{17.6 \pm 19.6}$ | $4.7 \pm 7.3$ | $7.9 \pm 10.9$ | - | $\mathbf{18.8 \pm 16.6}$ | $3.1 \pm 6.6$ | $3.7 \pm 8.3$ |

global model for each cluster of clients, resulting in $K$ global models. For personalized models like FedRep (Collins et al., 2021), or FedPer (Arivazhagan et al., 2019), we run the personalized algorithm on each cluster of clients. By running FL algorithms on clustered client, we ensure information exchange only between similar clients and improves the overall performance of federated learning algorithms by reducing the statistical dataset heterogeneity among clients.

We have run experiments on MNIST and CIFAR10 in which client datasets hold a clear cluster structure. We have also run experiments where there is no cluster structure in which clients are randomly assigned a pair of classes. In practice, we used the code of FedRep Collins et al. (2021) for the *FedAvg*, *FedRep* and *FedPer* and the spectral clustering method of scikit-learn (Pedregosa et al., 2011) (details are in Appendix D.7). Results are reported in Table 1 (with details in Appendix D.7). We can see that when there is a clear clustering structure among the clients, `FedWaD` is able to recover it and always improve the performance of the original federated learning algorithms. Depending on the algorithm, the improvement can be highly significant. For instance, for *FedRep*, the performance can be improved by 9 points for CIFAR10 and up to 29 for MNIST. Interestingly, even without clear clustering structure, `FedWaD` is able to almost always improve the performance of all federated learning algorithms (except for some specific cases of *FedPer*). Again for *FedRep*, the performance uplift can reach 19 points for CIFAR10 and 36 for MNIST. In terms of clustering, the "affinity" parameter of the spectral clustering algorithm seems to be the most efficient and robust one.

## 5 CONCLUSION

In this paper, we presented a principled approach for computing the Wasserstein distance between two distributions in a federated manner. Our proposed algorithm, called `FedWaD`, leverages the geometric properties of the Wasserstein distance and associated geodesics to estimate the distance while respecting the privacy of the samples stored on different devices. We established the convergence properties of `FedWaD` and provided empirical evidence of its practical effectiveness through simulations on various problems, including dataset distance and coreset computation. Our approach shows potential applications in the fields of machine learning and privacy-preserving data analysis, where computing distances for distributed data is a fundamental task.

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

# Appendices

## A  PROPERTY OF THE APPROXIMATING INTERPOLATING MEASURE

**Theorem 1.** *Assume that $\mu$ and $\xi^{(k)}$ are two discrete distributions with the same number of samples $n$ and uniform weights., Then for any $t$, the approximating interpolating measure given by equation Equation (10) is equal to the exact one Equation (5).*

*Proof.*  Remind that the approximating interpolating measure is defined as

$$\xi_t = \frac{1}{n}\sum_{i=1}^{n}\delta_{(1-t)x_i + tn(\mathbf{P}^\star \mathbf{X}')_i} \tag{11}$$

whereas the exact interpolating measure is defined as

$$\mu_t = \sum_{i,j}^{n,m}\mathbf{P}^\star_{i,j}\delta_{(1-t)x_i + tx'_j} \tag{12}$$

where $\mathbf{P}^\star$ is the optimal transportation plan between $\xi$ and $\xi'$, $x_i$ and $x'_j$ are the samples from these distributions and $\mathbf{X}'$ is the matrix of samples from $\xi'$. Because $\mu$ and $\xi^{(k)}$ have the same number of samples $n$ and uniform weights, $\mathbf{P}^\star$ is a weighted (by $1/n$) permutation matrix Peyré et al. (2019). Let us denote by $\sigma$ the permutation associated to $\mathbf{P}^\star$. Then, for the approximation, we have

$$\begin{aligned}
\xi_t &= \frac{1}{n}\sum_{i=1}^{n}\delta_{(1-t)x_i + tn(\mathbf{P}^\star \mathbf{X}')_i} \\
&= \frac{1}{n}\sum_{i=1}^{n}\delta_{(1-t)x_i + tx'_{\sigma(i)}} \\
&= \sum_{i=1}^{n}\frac{1}{n}\delta_{(1-t)x_i + tx'_{\sigma(i)}} \\
&= \mu_t
\end{aligned}$$

where the last equality comes from the fact that for each row $i$, $P^\star_{i,j}$ is non-zero only for $\sigma(i)$ column and $P^\star_{i,\sigma(i)} = 1/n$. $\qquad\square$

## B  PROOF OF THEOREM 2

**Theorem 2.** *Let $\mu$ and $\nu$ be two measures in $\mathscr{P}_p(X)$. For $k \in \mathbb{N}$, let $\xi^{(k)}_\mu$, $\xi^{(k)}_\nu$ and $\xi^{(k)}$ be interpolating measures computed at iteration $k$ as defined in Algorithm 1. Define*

$$A^{(k)} = \mathcal{W}_p(\mu, \xi^{(k)}_\mu) + \mathcal{W}_p(\xi^{(k)}_\mu, \xi^{(k)}) + \mathcal{W}_p(\xi^{(k)}, \xi^{(k)}_\nu) + \mathcal{W}_p(\xi^{(k)}_\nu, \nu)$$

*Then, the sequence $(A^{(k)})$ is non-increasing and converges to $\mathcal{W}_p(\mu, \nu)$.*

*Proof.*  First, remind that $\xi^{(k)}_\mu$ and $\xi^{(k)}_\nu$ are the interpolating measures between $\mu$ and $\xi^{(k-1)}$ and between $\xi^{(k-1)}$ and $\nu$ respectively, as defined in Algorithm 1. Likewise, $\xi^{(k+1)}_\mu$ and $\xi^{(k+1)}_\nu$ are interpolating measures between $\mu$ and $\xi^{(k)}$ and between $\xi^{(k)}$ and $\nu$ respectively. Hence, we have

$$\mathcal{W}_p(\mu, \xi^{(k+1)}_\mu) + \mathcal{W}_p(\xi^{(k+1)}_\mu, \xi^{(k)}) \leq \mathcal{W}_p(\mu, \xi^{(k)}_\mu) + \mathcal{W}_p(\xi^{(k)}_\mu, \xi^{(k)})$$

and

$$\mathcal{W}_p(\nu, \xi^{(k+1)}_\nu) + \mathcal{W}_p(\xi^{(k+1)}_\nu, \xi^{(k)}) \leq \mathcal{W}_p(\nu, \xi^{(k)}_\nu) + \mathcal{W}_p(\xi^{(k)}_\nu, \xi^{(k)})$$

These two inequalities lead to,

$$\mathcal{W}_p(\mu, \xi^{(k+1)}_\mu) + \mathcal{W}_p(\xi^{(k+1)}_\mu, \xi^{(k)}) + \mathcal{W}_p(\nu, \xi^{(k+1)}_\nu) + \mathcal{W}_p(\xi^{(k+1)}_\nu, \xi^{(k)})$$

$$\leq \mathcal{W}_p(\mu, \xi_\mu^{(k)}) + \mathcal{W}_p(\xi_\mu^{(k)}, \xi^{(k)}) + \mathcal{W}_p(\nu, \xi_\nu^{(k)}) + \mathcal{W}_p(\xi_\nu^{(k)}, \xi^{(k)})$$

Besides, since $\xi^{(k+1)}$ is an interpolating measure between $\xi_\mu^{(k+1)}$ and $\xi_\nu^{(k+1)}$, we have

$$\mathcal{W}_p(\xi_\mu^{(k+1)}, \xi^{(k+1)}) + \mathcal{W}_p(\xi^{(k+1)}, \xi_\nu^{(k+1)}) \leq \mathcal{W}_p(\xi_\mu^{(k+1)}, \xi^{(k)}) + \mathcal{W}_p(\xi^{(k)}, \xi_\nu^{(k+1)})$$

and

$$A^{(k+1)} \leq A^{(k)}$$

Hence, the sequence $(A^{(k)})$ is non-increasing. Additionally, by the triangle inequality, we have for any $k \in \mathbb{N}$,

$$\mathcal{W}_p(\mu, \nu) \leq A^{(k)}$$

We conclude by using the monotone convergence theorem: since $(A^{(k)})$ is non-increasing and bounded sequence below, then it converges to its infimum.

We now justify why the limit of $(A^{(k)})$ is $\mathcal{W}_p(\mu, \nu)$. At convergence, we have reached a stationary point in the $(A^{(k)})$,

$$\lim_{k \to +\infty} A^{(k)} = \mathcal{W}_p(\mu, \xi_\mu^{(\infty)}) + \mathcal{W}_p(\xi_\mu^{(\infty)}, \xi^{(\infty)}) + \mathcal{W}_p(\xi^{(\infty)}, \xi_\nu^{(\infty)}) + \mathcal{W}_p(\xi_\nu^{(\infty)}, \nu)$$

and there are an infinite number of triplets $(\xi_\mu^{(\infty)}, \xi_\nu^{(\infty)}, \xi^{(\infty)})$ that allow to reach this value $A^{(\infty)}$ by the nature of the algorithm. By definition, $\xi_\mu^{(\infty)}$ and $\xi_\nu^{(\infty)}$ are interpolating measures between $\mu$ and $\xi^{(\infty)}$ and between $\xi^{(\infty)}$ and $\nu$ respectively. At convergence, $(\xi_\mu^{(\infty)}, \xi_\nu^{(\infty)}, \xi^{(\infty)})$ are fixed points of the algorithm, and we show here that $\xi^{(\infty)}$ is an interpolating measure of $\mu$ and $\nu$ in addition to be an interpolating measure of $\xi_\mu^{(\infty)}$ and $\xi_\nu^{(\infty)}$. For any $\xi^{(\infty)}$, $\xi_\mu^{(\infty)}$ can be chosen as any interpolating measure between $\mu$ and $\xi^{(\infty)}$. The same reasoning holds for $\xi_\nu^{(\infty)}$ and $\nu$. Then since $\xi^{(\infty)}$ is an interpolating measure of $\xi_\mu^{(\infty)}$ and $\xi_\nu^{(\infty)}$ and $\mu$ and $\nu$ are possible choices of interpolating measures, it yields that $\xi^{(\infty)}$ is indeed an interpolating measure of $\mu$ and $\nu$. Hence, we have

$$\lim_{k \to +\infty} A^{(k)} = \mathcal{W}_p(\mu, \xi_\mu^{(\infty)}) + \mathcal{W}_p(\xi_\mu^{(\infty)}, \xi^{(\infty)}) + \mathcal{W}_p(\xi^{(\infty)}, \xi_\nu^{(\infty)}) + \mathcal{W}_p(\xi_\nu^{(\infty)}, \nu)$$
$$= \mathcal{W}_p(\mu, \xi^{(\infty)}) + \mathcal{W}_p(\xi^{(\infty)}, \nu)$$
$$= \mathcal{W}_p(\mu, \nu)$$

where the first equality results from the fact that $\xi_\mu^{(\infty)}$ and $\xi_\nu^{(\infty)}$ are interpolating measures between $\mu$ and $\xi^{(\infty)}$ and between $\xi^{(\infty)}$ and $\nu$ respectively and the second equality is obtained from the fact that $\xi^{(\infty)}$ is also an interpolating measure between $\mu$ and $\nu$ as belonging to the geodesic between $\mu$ and $\nu$. □

## C  CONVERGENCE RATE OF THE ALGORITHM FOR GAUSSIAN DISTRIBUTIONS WITH SAME COVARIANCE

In this section, we show that when $\mu$ and $\nu$ are Gaussians after one iteration, we can infer $W(\mu, nu)$ and the sequence of iterates $(\xi^{(k)})$ obtained for $t = 0.5$ converges to the $\xi^\star$ the interpolating measure between $\mu$ and $\nu$ for $t = 0.5$

**Theorem 3.** *Assume that $\mu$, $\nu$ and $\xi^{(0)}$ are three Gaussian distributions with the same covariance matrix $\Sigma$ ie $\mu \sim \mathcal{N}(\mathbf{m}_\mu, \Sigma)$, $\nu \sim \mathcal{N}(\mathbf{m}_\nu, \Sigma)$ and $\xi^{(0)} \sim \mathcal{N}(\mathbf{m}_{\xi^{(0)}}, \Sigma)$. Further assume that we are not in the trivial case where $\mathbf{m}_\mu$, $\mathbf{m}_\nu$, and $\mathbf{m}_{\xi^{(0)}}$ are aligned. Applying our algorithm Algorithm 1 with $t = 0.5$ and the squared Euclidean cost, we have the following properties:*

1. *all interpolating measures $\xi_\mu^{(k)}, \xi_\nu^{(k)}, \xi^{(k)}$ are isotropic Gaussian distributions with the same covariance matrix $\Sigma$*

2. *for any $k \geq 1$, $\mathcal{W}_2(\mu, \nu) = \|\mathbf{m}_\mu - \mathbf{m}_\nu\|_2 = 2\|\mathbf{m}_{\xi_\mu^{(k)}} - \mathbf{m}_{\xi_\nu^{(k)}}\|_2$*

3. *$\mathcal{W}_2(\xi^{(k)}, \xi^\star) = \frac{1}{2}\mathcal{W}_2(\xi^{(k-1)}, \xi^\star)$*

4. $\mathcal{W}_2(\mu, \xi^{(k)}) + \mathcal{W}_2(\xi^{(k)}, \nu) - \mathcal{W}_2(\mu, \nu) \leq \frac{1}{2^{k-1}} \mathcal{W}_2(\xi^{(0)}, \xi^{(\star)})$

*Proof.* The first point comes from the fact that Wasserstein barycenter of Gaussians are Gaussians Agueh & Carlier (2011); Peyré et al. (2019). For isotropic Gaussians with same covariance, the covariance matrice of the barycenter remains unchanged while the mean is the barycenter mean. So, in our case, the interpolating measure with $t = 0.5$ *i.e* the uniform barycenter of two measures, say $\mu$ and $\xi^{(k-1)}$, is $\xi_\mu^{(k)} \sim \mathcal{N}(\mathbf{m}_{\xi_\mu^{(k)}}, \Sigma)$, where $\mathbf{m}_{\xi_\mu^{(k)}} = \frac{1}{2}(\mathbf{m}_\mu + \mathbf{m}_{\xi^{(k-1)}})$. The consequence of this first point of the theorem is that since we are going to deal with same covariance Gaussian distributions, then the Wasserstein distance between any pair of measures involved in our algorithm only depends on the Euclidean distance of their means and we will use interchangeably the Euclidean distance and the Wasserstein distance.

The second point is proven by using geometrical arguments in the plane $(P)$ in which the three points, for $k \geq 1$, $\mathbf{m}_\mu, \mathbf{m}_\nu, \mathbf{m}_{\xi^{(k-1)}}$ lie (note that based on our assumption, this plane always exists). By definition of $\xi_\mu^{(k)}$ and $\xi_\nu^{(k)}$ and given the above point, we have

$$\mathbf{m}_{\xi_\mu^{(k)}} = \frac{1}{2}(\mathbf{m}_\mu + \mathbf{m}_{\xi^{(k-1)}}) \quad \text{and} \quad \mathbf{m}_{\xi_\nu^{(k)}} = \frac{1}{2}(\mathbf{m}_\nu + \mathbf{m}_{\xi^{(k-1)}})$$

By using the intercept theorem, since $t = \frac{1}{2}$, in the plane $(P)$, the segment $[\mathbf{m}_{\xi_\mu^{(k)}}, \mathbf{m}_{\xi_\nu^{(k)}}]$ is parallel to the segment $[\mathbf{m}_\mu, \mathbf{m}_\nu]$ and we have :

$$\frac{1}{2} = \frac{\|\mathbf{m}_{\xi_\mu^{(k)}} - \mathbf{m}_{\xi^{(k-1)}}\|_2}{\|\mathbf{m}_\mu - \mathbf{m}_{\xi^{(k-1)}}\|_2} = \frac{\|\mathbf{m}_{\xi_\nu^{(k)}} - \mathbf{m}_{\xi^{(k-1)}}\|_2}{\|\mathbf{m}_\nu - \mathbf{m}_{\xi^{(k-1)}}\|_2} == \frac{\|\mathbf{m}_{\xi_\nu^{(k)}} - \mathbf{m}_{\xi_\mu^{(k)}}\|_2}{\|\mathbf{m}_\nu - \mathbf{m}_\mu\|_2}$$

which gives us the second point.

For the third point, we are going to consider geometrical arguments similar as above. However, we are going to first show that for a given $k$, the mid point, denoted as $\hat{\xi}^{(k)}$, between $\xi^{(k-1)}$ and $\xi^\star$ is also $\xi^{(k)}$ as defined by our algorithm.

By definition, $\xi^\star$ is the mid point interpolating measure between $\mu$ and $\nu$, whose mean is $\frac{1}{2}(\mathbf{m}_\mu + \mathbf{m}_\nu)$. Since $\hat{\xi}^{(k)}$ and $\xi_\mu^{(k)}$ are respectively the mid point measure between $\xi^{(k-1)}$ and $\xi^\star$ and $\mu$ and $\xi^{(k-1)}$, we can apply the intercept theorem in the appropriate plane and get

$$W_2(\hat{\xi}^{(k)}, \xi_\mu^{(k)}) = \frac{1}{2} W_2(\mu, \xi^\star)$$

Using a similar reasoning using $\nu$, we get

$$W_2(\hat{\xi}^{(k)}, \xi_\nu^{(k)}) = \frac{1}{2} W_2(\nu, \xi^\star)$$

Summing these two equations, we obtain

$$W_2(\hat{\xi}^{(k)}, \xi_\mu^{(k)}) + W_2(\hat{\xi}^{(k)}, \xi_\nu^{(k)}) = \frac{1}{2} W_2(\mu, \xi^\star) + \frac{1}{2} W_2(\nu, \xi^\star) = \frac{1}{2} W_2(\mu, \nu) = W_2(\xi_\mu^{(k)}, \xi_\nu^{(k)})$$

where the second equality comes from the fact that $\xi^\star$ is an interpolant measure of $\mu$ and $\nu$, while the last equality comes from the second point of the theorem.

Hence, since we have $W_2(\hat{\xi}^{(k)}, \xi_\mu^{(k)}) + W_2(\hat{\xi}^{(k)}, \xi_\nu^{(k)}) = W_2(\xi_\mu^{(k)}, \xi_\nu^{(k)})$, it also mean than

$$\hat{\xi}^{(k)} \in \arg\min_\xi \frac{1}{2} W_2(\xi_\mu^{(k)}, \xi) + \frac{1}{2} W_2(\xi, \xi_\nu^{(k)})$$

and $\hat{\xi}^{(k)}$ is also the midpoint interpolating measure between $\xi_\mu^{(k)}$ and $\xi_\nu^{(k)}$.

Then, applying the intercept theorem with $\xi^{(k-1)}, \xi^{(k)}, \xi^\star, \mu$ and $\xi_\mu^{(k)}$, we obtain the desired result

$$\frac{1}{2} W_2(\xi^{(k-1)}, \xi^\star) = W_2(\xi^{(k)}, \xi^\star)$$

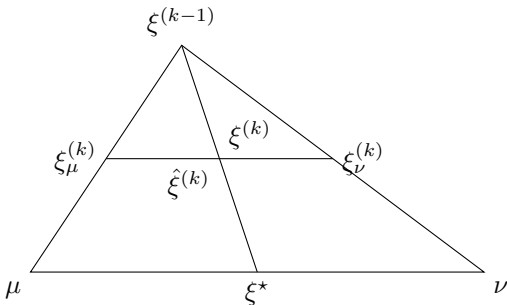

Figure 7: Illustration of the geometrical interpretation of the algorithm and its convergence proof for Gaussian distributions with same covariance, based on the intercept theorem.

Finally, given all the above, it is simple to show the convergence rate of the algorithm using simple triangle inequalities.

$$
\begin{aligned}
\mathcal{W}_2(\mu, \xi^{(k)}) + \mathcal{W}_2(\xi^{(k)}, \nu) - \mathcal{W}_2(\mu, \nu) &\leq \mathcal{W}_2(\mu, \xi^\star) + \mathcal{W}_2(\xi^\star, \xi^{(k)}) + \mathcal{W}_2(, \xi^{(k)}, \xi^\star) + \mathcal{W}_2(\xi^\star, \nu) - \mathcal{W}_2(\mu, \nu) \\
&= 2\mathcal{W}_2(\xi^{(k)}, \xi^\star) \\
&= \frac{1}{2^{k-1}} \mathcal{W}_2(\xi^{(0)}, \xi^\star)
\end{aligned}
$$

$\square$

## D    ADDITIONAL EXPERIMENTS

### D.1    TOY ANALYSIS : THE IMPACT OF APPROXIMATING THE INTERPOLATING MEASURE

We propose to analyze in this section the benefits and disadvantages of approximating the interpolating measure instead of using the exact one as given in Equation (5). For this purpose, we compare the running time and the accuracy of the exact Wasserstein distance, our exact FedWaD, and our approximate FedWaD for estimating the Wasserstein distance between two Gaussians distributions. The Gaussians are different means but same covariances so that the true Wasserstein distance is known and equal to the Euclidean distance between the means. We have considered two different settings ($d = 2$ and $d = 50$) of Gaussians dimensionality. For the first case ($d = 2$), we detail the results presented in the main paper. Note that when the dimensionality of the Gaussians are set to 50, we do not expect the Wasserstein distance nor `FedWaD` to provide a good estimation of the closed form distance between these distributions, due to the curse of dimensionality of the Wasserstein distance (Fournier & Guillin, 2015)

As default parameter for our approximate FedWaD, we considered 20 iterations and a support of size 10, then we varied the number of samples $n$ from 10 to 10000. We have run experiments in different settings

- we analyzed the impact of sample ratio between the two distributions, as this may impact the support size of the approximating interpolating measure accross `FedWaD` iterations.
- we made varying the support size of the approximating interpolating measure at fixed sample ratio.

Results have been averaged over 10 runs.

**Analyzing the impact of sample ratio**    Given the setting with uniform weights, when the sample ratio is 1, the optimal plan is theoretically a scaled permutation matrix. Hence, the support size of the exact interpolating measure is expected, in theory, to be fixed and equal to $N$. When the ratio of samples is different to 1, the support size of the exact interpolating may increase at each iteration of the algorithm and leads to a larger running time. Figure 8 - left panel - shows the running time of

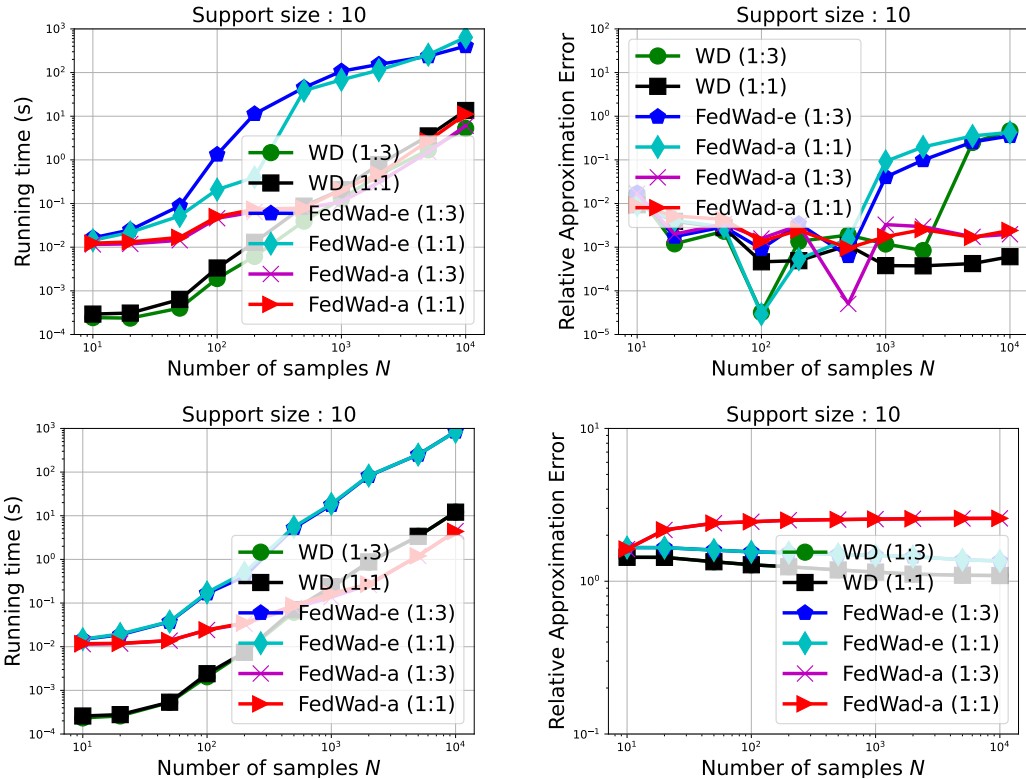

Figure 8: For different sample ratios, (1:3) or (1:1), in the two distributions we report the performance of the different models. For our approximated FedWaD, we have set the support size to 10. (top) $d = 2$ (bottom) $d = 50$. (left) running time. (right) relative error.

all compared methods as well as their relative error - right panel - compared to the true Wasserstein distance. We note that for $2d$ Gaussians, both the Wasserstein distance and our approximated FedWaD with support size of 10 the running time is increasing with a natural computational overhead for the 1:1 sample ratio (as we have more samples). For the exact FedWaD, the behavior is different. the running time for the 1:3 sample ratio is larger than the 1:1. This is due to the optimal transportation plan $\mathbf{P}^\star$ not being exactly a scaled permutation matrix. As a result, the support size of the interpolating measure increases with the number of samples, leading to computational overhead for the method. For $50d$ Gaussians, the differences in running time between the different sample ratio are negligible.

In the case of $2d$ Gaussians (top row), For the relative error, for $N < 1000$, we note that all methods achieve similar errors. Numerical errors start to appear for exact FedWaD and the Wasserstein distance for respectively $N \geq 1000$ and $N \geq 5000$ depending on the sample ratio. Interestingly, the approximated FedWaD is robust to large number of samples and achieves similar errors as for small number of samples. For higher dimensions (bottom row), all the methods are not able to provide accurate estimation of the Wasserstein distance and with the worst relative error for the approximated FedWaD with a support size of 10. Nonetheless, we want to emphasize that despite this lack of accuracy, the approximated `FedWaD` can be useful in high-dimension problems as we have shown for the other experiments.

**Analyzing the support size of approximated interpolating measure**   Figure 9 shows the running time and the relative error of the different methods for a sample ratio of $1 : 3$ and when the support sizes of the approximating interpolating measure are 2,10 or 100. We clearly remark the computational cost of a larger support size with a benefit in terms of approximation error appearing mostly when $N \geq 1000$ and for small dimension problems (top row). For higher dimension problems (bottom row), we see again the benefit on running time of the approximated approach, yet with a larger approximation error.

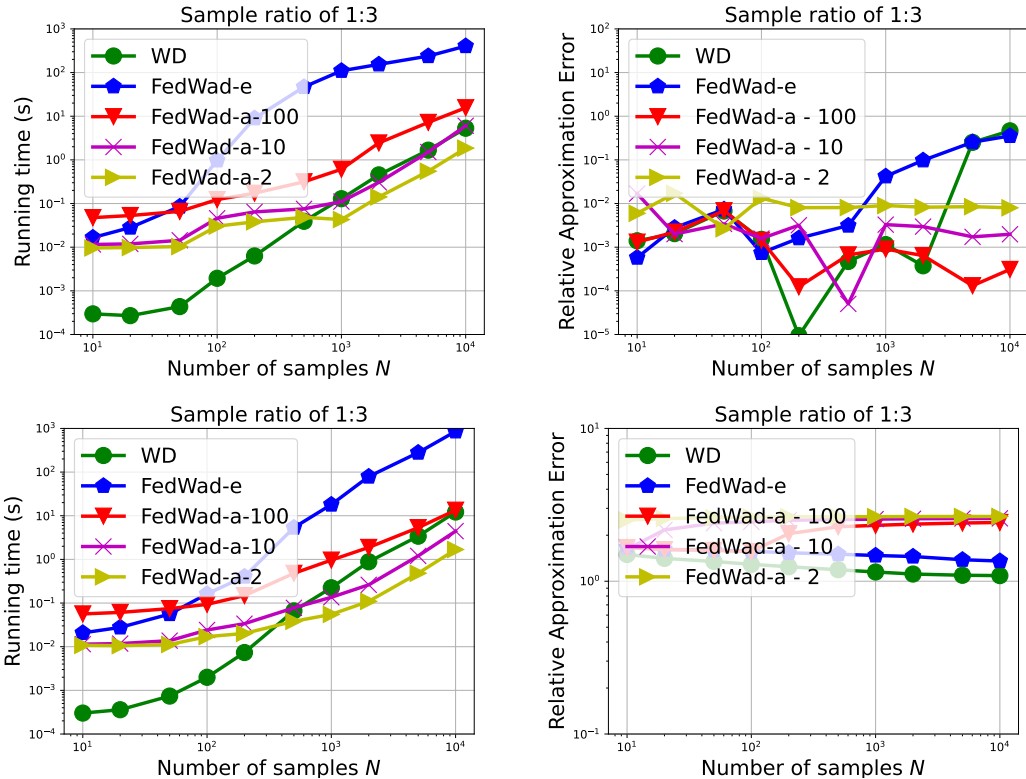

Figure 9: For increasing number of samples, we report (top) $d = 2$ (bottom) $d = 50$. (left) Running time of the Wasserstein distance, our exact FedWaD and our approximate FedWaD. (right) the relative error of the different models : the computed Wasserstein distance, our exact FedWaD and the approximated FedWad with a support size of $10$ and $100$. The first distribution has a number of samples $N$ and the second ones $N/3$.

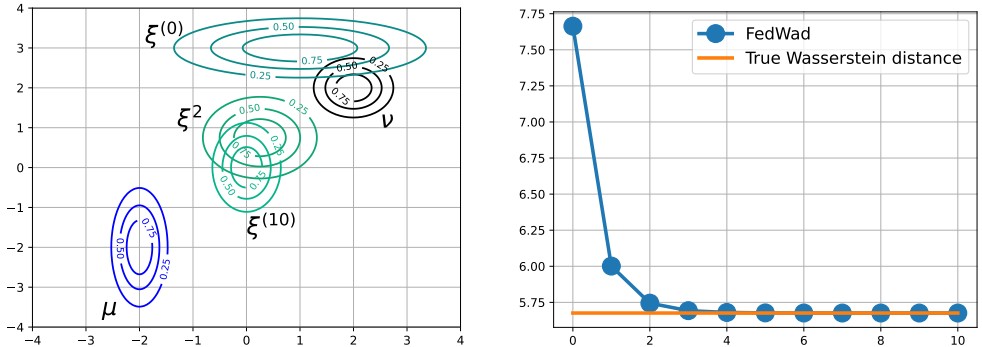

Figure 10: We illustrate here how our algorithm behaves when the distributions are continuous. (left) we plot the distributions $\mu$ and $\nu$ as well as the interpolating measure $\xi^{(k)}$ (right) we plot the evolution of the Wasserstein distance between $\mu$ and $\nu$ as computed by FedWad.

## D.2 TOY ANALYSIS : CONTINUOUS DISTRIBUTIONS

Our algorithm can be applied to continuous distributions as long as it is possible to compute an element of the geodesic between the two distributions. For multivariate Gaussian distributions, the transport map exists and elements of the geodesics are well-defined. However, closed-form of the mean and the covariance matrix of interpolating measures are not available except when the covariance

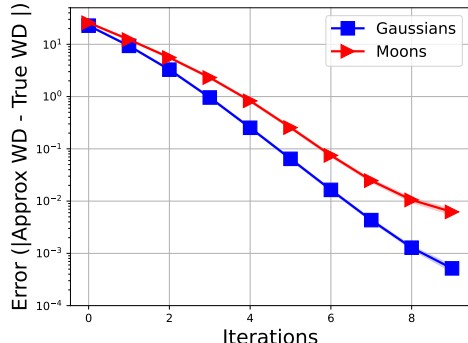

Figure 11: Example of convergence of FedWad when computing the distance between two Gaussian distributions and between two moon-shaped distributions.

matrices between $\mu$ and $\nu$ are jointly diagonalizable. Hence, as an example, we have applied our algorithm for two continuous Gaussians distributions $\mu \sim \mathcal{N}(\mathbf{m}_\mu, \Sigma_\mu)$ and $\nu \sim \mathcal{N}(\mathbf{m}_\nu, \Sigma_\nu)$ where $\Sigma_\mu$ and $\Sigma_\nu$ are diagonal matrices. $\xi^{(0)}$ is also defined as a diagonal Gaussian distribution. Mean and covariance of an interpolating measures for $t = 0.5$ are computed as follows (say between $\mu$ and $\nu$):

$$\mathbf{m} = \frac{1}{2}(\mathbf{m}_\mu + \mathbf{m}_\nu)$$

$$\Sigma = \left(\frac{1}{2}\Sigma_\mu^{1/2} + \frac{1}{2}\Sigma_\nu^{1/2}\right)^2$$

Figure 10 shows an example of the evolution of the Wasserstein distance between $\mu$ and $\nu$ as computed by our algorithm as well as the interpolating measure $\xi^{(k)}$ for different values of $k$. We can see that the Wasserstein distance converges to the true Wasserstein distance between $\mu$ and $\nu$ in about 10 iterations confirming the linear convergence rate.

### D.3 COMPARING CONVERGENCE RATE

In order to gain an insight about the convergence rate of our algorithm for non-Gaussian distributions, we have compared how fast FedWaD converges to the true Wasserstein distance when comparing two 2D Gaussians and when comparing two 2D moon-shaped distributions. We have considered 200 samples per distribution and computed the exact FedWaD using 10 iterations. Figure 11 shows the evolution of the Wasserstein distance between the two distributions as a function of the number of iterations. We can see that the convergence rate for the two moon-shaped distributions is slower than the ones of the Gaussians, which is about 1.5 order of magnitude, and it tends to decrease as iterations increase.

### D.4 DETAILS ON CORESET AND ADDITONAL RESULTS

**Experimental setting** We sampled 20000 examples randomly from the `MNIST` dataset, and dispatched them at random on 100 clients but such that only a subset $K$ of the 10 classes is present on each client. We learn 10 coresets over 1000 epochs and at each epoch, we assume that only 10 random clients are available and can be used for computing `FedWaD`. For `FedWaD`, the support size of the interpolating measure has been set to either 10 or 100 and the number of iteration in `FedWaD` to 20.

We have reproduced in here the same MNIST experiment (which results are reproduced in Figure 12) on coreset for the `FashionMNIST` dataset, and we can notice, in Figure 13 that we obtain similar results as for the `MNIST` dataset. When the number of shared classes $K$ is large enough, the coreset is not able to capture the different modes in the dataset. And again, we remark that the support size of the approximate interpolating measure has few impacts on the result. For both datasets, the loss landscape of the coreset learning reveals that our `FedWaD`-based approaches yield to a worse minimum than the exact Wasserstein distance, which is mostly due to the interpolating measure approximation. Figure 14 plots the performance of a nearest neighbor classifier based on the coresets

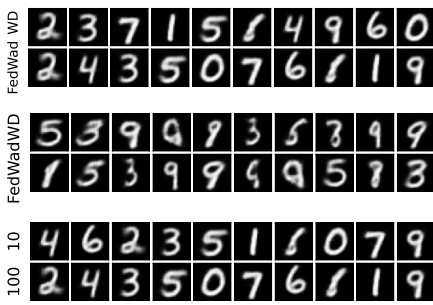 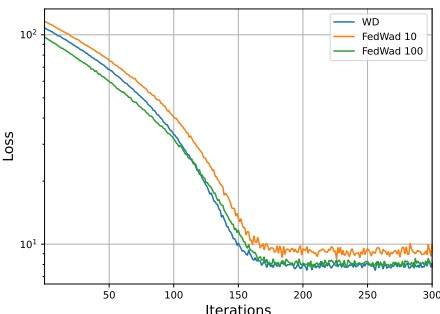

Figure 12: Examples of the 10 coreset obtained with for each panel *(top-row)* the exact Wasserstein and *(bottow-row)* `FedWaD` for the `MNIST` dataset. Different panels correspond to different number of classes $K$ on each client: *(top)* $K = 8$, *(middle)* $K = 2$, *(bottom)* support of the interpolating measure varying from 10 to 100. As class diversity on each client increases, the coreset is less effective at capturing the 10 modes of the dataset

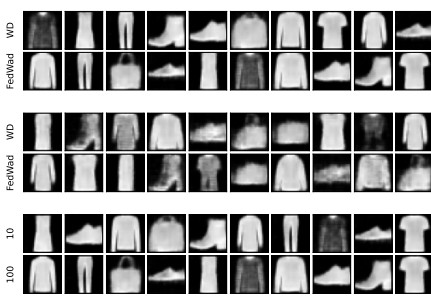 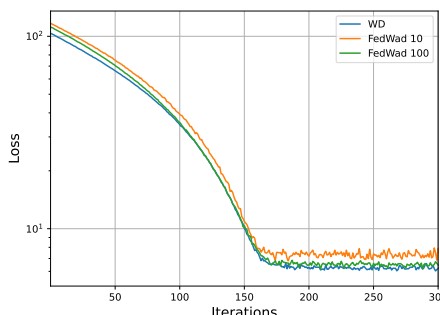

Figure 13: Examples of the 10 coreset obtained with for each panel *(top-row)* the exact Wasserstein and *(bottow-row)*, our FedWaD for the `FashionMNIST` dataset. Different panels correspond to different number of classes $K$ on each client: *(top)* $K = 8$, *(middle)* $K = 2$, *(bottom)* support of the interpolating measure for $K = 8$.

learnt from each client for varying number of clients. Results show that coreset-based approaches are competitive, especially for high number of clients, with personalized FL algorithms, which are known to be the best performing FL algorithms in practice.

### D.5 DETAILS ON FEDERATED OTDD EXPERIMENTS

**Geometric dataset distances via federated Wasserstein distance.** Transfer learning and domain adaptation are important ML paradigms, which aim at transferring knowledge across similar domains. The main underlying concept in these approaches is the notion of distance or similarity between datasets. Transferring knowledge between comparable domains is typically simpler than between distant ones. In certain applications, it is relevant to find datasets from which one can transfer knowledge from without disclosing the target dataset. This may be the case, for instance, in applications with low-resource clients storing sensitive data. In this case, the practitioner may want to find a dataset similar enough to the client's dataset, in order to transfer knowledge from it. In practice, a server would train a classifier on a dataset that is similar to the client dataset, and the client would then use this classifier to perform inference on its own data.

In that context, our goal is to propose a distance between datasets that can be computed in a federated way based on `FedWaD`. We leverage the distance proposed in Alvarez-Melis & Fusi (2020), which is based on the Wasserstein distance between two labeled datasets $\mathcal{D}$ and $\mathcal{D}'$. The ground metric is defined by,

$$d_{\mathcal{D}}((x, y), (x', y')) \triangleq (d(x, x') + \mathcal{W}_2^2(\alpha_y, \alpha'_y))^{1/2} \tag{13}$$

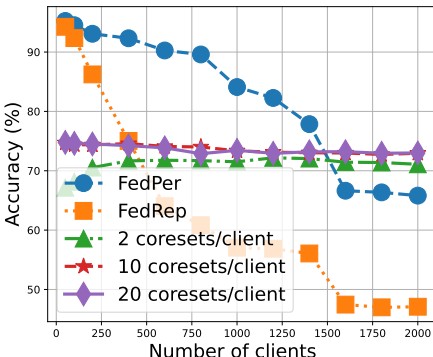

Figure 14: FashionMNIST performance of a nearest neighbor classifier based on the coresets learnt from each client for varying number of clients and number of coresets per clients We have compared to the performance of two personalized FL algorithms.

where $d$ is a distance between two features $x$ and $x'$, and $\alpha_y$ is the class-conditional distribution of $x$ given $y$. In order to reduce computational complexity, Alvarez-Melis & Fusi (2020) assume the class-conditionals are Gaussian, so that $\mathcal{W}_2$ boils down the 2-Bures-Wasserstein distance, which is available in closed form:

$$\mathcal{W}_2^2(\alpha_y, \alpha_{y'}) = \|m_y - m_{y'}\|_2^2 + \|\Sigma_y - \Sigma_{y'}\|_F^2 \tag{14}$$

where $m_z$ and $\Sigma_z$ denote the mean and covariance of $\alpha_z$.

`FedWaD` needs vectorial representations of the data to compute intermediate measures. The Bures-Wasserstein distance allows us to conveniently represent $\alpha_y$ as the concatenation of the mean $m_y$ and vectorized covariance $\Sigma_y$. Hence, we can compute the distance between two datasets $\mathcal{D}$ and $\mathcal{D}'$ by augmenting each example from those datasets with the corresponding class-conditional mean and vectorized covariance, and using the $\ell_2$ norm as the ground metric in the Wasserstein distance. One can eventually reduce the dimension the augmented representation by considering only the diagonal of the covariance matrix.

### D.6 FEDERATED OTDD ANALYSIS

To evaluate our procedure, we replicated the experiments of Alvarez-Melis & Fusi (2020) on the dataset selection for transfer learning: given a source dataset, the goal is to find a target one which is the most similar to the source. We considered four real datasets, namely `MNIST`, `KMNIST`, `USPS` and `FashionMNIST`. We first analyze the impact of two hyperparameters, the number of epochs and the number of support points in the interpolating measure, on the distance computation between $5000$ samples from `MNIST` and `KMNIST`, Figure 15 shows the evolution of the distance between `MNIST` and `KMNIST` as well as the running time for varying values of hyperparameters. The number of epochs has a very small impact on the distance and using $10$ epochs suffices to get a reasonably accurate approximation of the distance. On the other hand, the number of support point seems more critical, and we need at least $5000$ support points to obtain a very accurate approximation, although we have a nice linear convergence of the distance with respect to support size.

We also analyzed the impact of the dataset size on the distance computation and running time: Figure 16 shows the evolution of the distance and the running time with respect to the the sample size in the two distributions. We note that the order relation is preserved between the two distances for all possible range of sample size. Another interesting observation is that as long as the sample size is smaller than the support size of the interpolating measure, `FedWaD` provides an accurate estimation of the distance. When the sample size is larger then the distance is overestimated. This is due to a less accurate estimation of an exact interpolating measure (which is supported on $2n + 1$ points). Regarding computational efficiency, we observe that for small support size of the interpolating measure, the running time increases at the same rate as the sample size, whereas for larger support size, the running time increases 10-fold for an 100-fold increase in sample size.

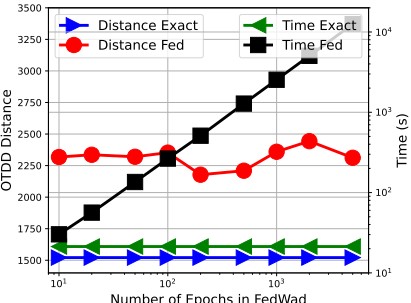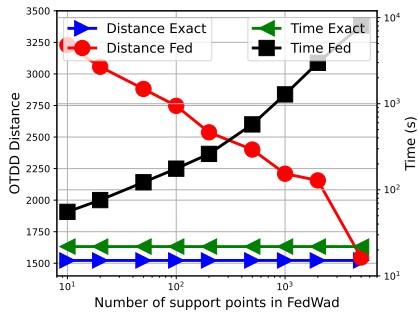

Figure 15: `FedWaD` and OTDD distances on `MNIST-KMNIST` and its running time against *(left)* the number of epochs and *(right)* the number of support points in the interpolating measure. For each plot, the left and right $y$-axis report the distance and the running time respectively.

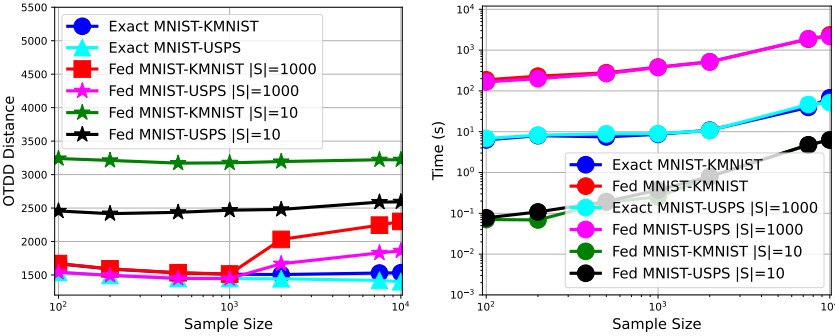

Figure 16: *(left)* Distance and *(right)* running time against the dataset size for the `MNIST-KMNIST` an `MNIST-USPS` distances, for varying number of support points $|S|$

### D.7 BOOSTING FL METHODS

We provide here more detailed results about our experiments on boosting FL methods. Figure 17 shows the distance matrices obtained for MNIST and CIFAR10 when the number of clients is 20 for different structures on the clients datasets. We can clearly see the cluster structure on the MNIST dataset when it exists, but when there is no structure, the distance matrix is more uniform yet show some variations For CIFAR10, no clear structure is visible on the distance matrix as the dataset is more complex. Nonetheless, our experiments on boosting FL methods show that even in this case, clustering c lients can help improve the performances of federated learning algorithms.

Those distance matrices are the one we use as the input of the spectral clustering algorithm. We used the spectral clustering algorithm of scikit-learn (Pedregosa et al., 2011) with the following setting::

- we denoted as "affinity", the setting in which the distance matrix, after rescaling, is used as affinity matrix, where larger values indicate greater similarity between instances. (see affinity parameter set to 'precomputed' In scikit-learn)

- we denote as Sparse G. (3) and Sparse G. (5) the setting in which the distance matrix is interpreted as a sparse graph of distances, and construct a binary affinity matrix from the (3 or 5) nearest neighbors of each instance. matrix is computed

**Details on the cluster structure** We have built this cluster structure on the client datasets by assigning to each client one pair of classes among the following 5 ones : $[(0,1),(2,3),(4,5),(6,7),(8,9)]$. When the number of clients in equal to 10, each cluster is composed of 2 clients. For a larger number of clients, each cluster is of random size with a minimum of 2 clients.

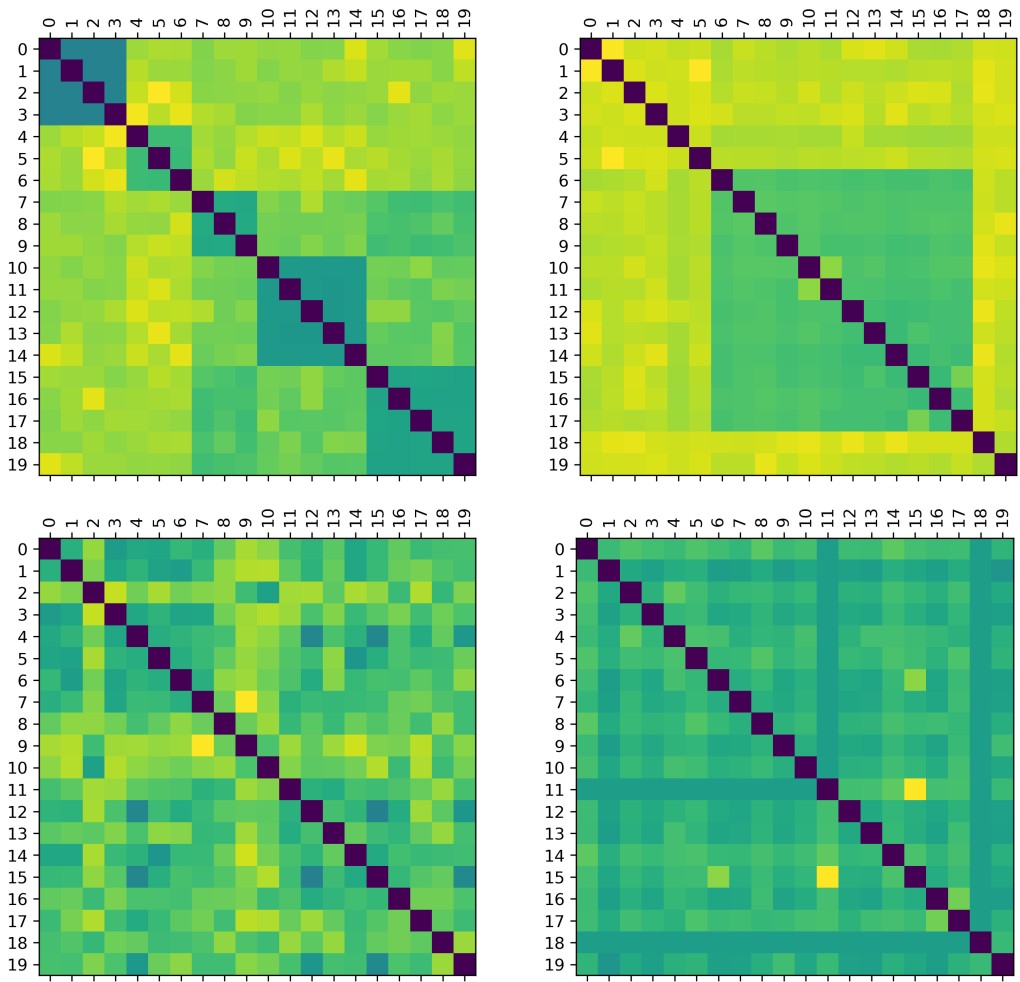

Figure 17: (left) MNIST and (right) CIFAR10 distance matrices for 20 clients computed using our Federated OTDD . On the top row, we have imposed a cluster structure on client datasets while on the bottom row, there is no specific sructure. We can note that this structure is clearly visible on the MNIST dataset but less on CIFAR10. Eventhough, clustering clients will help improve federated learning algorithm performances.

**Practical algorithmic details** In practice, we used the code of FedRep Collins et al. (2021) for the *FedAvg*, *FedRep* and *FedPer* and the spectral clustering method of scikit-learn Pedregosa et al. (2011). The federated OT distance dataset has been computed on the original data space while for CIFAR10, we have worked on the 784-dimensional code obtained from an (untrained) randomly initialized autoencoder. We have also considered the case where the there is no specific clustering structure on the clients as they randomly select a pair of classes among the 10 ones.

**Extra results** Performance results on federated learning are reported below for different settings. Table 2 and Table 3 show the results for MNIST respectively with and without client structure. Table 4 and Table 5 report similar results for CIFAR10.

| | | Clustered | | | | | |
| | | Affinity | | Sparse G. (3) | | Sparse G. (5) | |
| | Vanilla | 10 | 100 | 10 | 100 | 10 | 100 |
|---|---|---|---|---|---|---|---|
| FedAvg | | | | | | | |
| 10 | $19.6 \pm 0.9$ | $\mathbf{99.6 \pm 0.0}$ | $\mathbf{99.6 \pm 0.0}$ | $90.5 \pm 8.7$ | $91.8 \pm 9.6$ | $84.5 \pm 8.0$ | $85.2 \pm 5.7$ |
| 20 | $26.3 \pm 3.8$ | $\mathbf{99.5 \pm 0.0}$ | $\mathbf{99.5 \pm 0.0}$ | $\mathbf{99.5 \pm 0.0}$ | $\mathbf{99.5 \pm 0.0}$ | $91.5 \pm 10.3$ | $96.5 \pm 6.0$ |
| 40 | $39.1 \pm 9.0$ | $\mathbf{99.2 \pm 0.1}$ | $\mathbf{99.2 \pm 0.1}$ | $91.1 \pm 6.5$ | $\mathbf{99.2 \pm 0.1}$ | $94.5 \pm 9.4$ | $\mathbf{99.2 \pm 0.1}$ |
| 100 | $39.2 \pm 7.7$ | $\mathbf{98.9 \pm 0.0}$ | $\mathbf{98.9 \pm 0.0}$ | $95.9 \pm 4.6$ | $96.7 \pm 3.8$ | $98.4 \pm 0.8$ | $\mathbf{98.9 \pm 0.0}$ |
| FedRep | | | | | | | |
| 10 | $71.6 \pm 10.5$ | $\mathbf{99.4 \pm 0.0}$ | $\mathbf{99.4 \pm 0.1}$ | $94.3 \pm 7.7$ | $99.0 \pm 0.5$ | $95.5 \pm 5.6$ | $90.5 \pm 6.6$ |
| 20 | $81.1 \pm 8.1$ | $\mathbf{99.1 \pm 0.0}$ | $\mathbf{99.1 \pm 0.1}$ | $\mathbf{99.1 \pm 0.0}$ | $\mathbf{99.1 \pm 0.0}$ | $98.2 \pm 1.3$ | $99.0 \pm 0.2$ |
| 40 | $88.8 \pm 10.4$ | $98.9 \pm 0.1$ | $98.9 \pm 0.0$ | $93.3 \pm 7.1$ | $99.0 \pm 0.1$ | $96.7 \pm 4.5$ | $\mathbf{99.0 \pm 0.1}$ |
| 100 | $93.0 \pm 3.9$ | $\mathbf{98.6 \pm 0.1}$ | $\mathbf{98.6 \pm 0.1}$ | $98.4 \pm 0.1$ | $98.4 \pm 0.1$ | $98.5 \pm 0.1$ | $98.5 \pm 0.1$ |
| FedPer | | | | | | | |
| 10 | $86.7 \pm 4.3$ | $\mathbf{99.6 \pm 0.0}$ | $\mathbf{99.6 \pm 0.0}$ | $99.5 \pm 0.1$ | $\mathbf{99.6 \pm 0.1}$ | $98.4 \pm 2.0$ | $98.9 \pm 1.0$ |
| 20 | $94.3 \pm 4.3$ | $\mathbf{99.5 \pm 0.0}$ | $\mathbf{99.5 \pm 0.0}$ | $\mathbf{99.5 \pm 0.0}$ | $\mathbf{99.5 \pm 0.0}$ | $99.3 \pm 0.3$ | $\mathbf{99.5 \pm 0.0}$ |
| 40 | $94.7 \pm 7.6$ | $\mathbf{99.2 \pm 0.1}$ | $\mathbf{99.2 \pm 0.1}$ | $99.1 \pm 0.2$ | $\mathbf{99.2 \pm 0.1}$ | $97.9 \pm 2.7$ | $\mathbf{99.2 \pm 0.1}$ |
| 100 | $98.1 \pm 0.1$ | $\mathbf{98.9 \pm 0.0}$ | $\mathbf{98.9 \pm 0.0}$ | $98.8 \pm 0.2$ | $98.8 \pm 0.1$ | $\mathbf{98.9 \pm 0.0}$ | $\mathbf{98.9 \pm 0.0}$ |
| Average Uplift | - | $29.8 \pm 28.4$ | $29.8 \pm 28.4$ | $27.2 \pm 26.6$ | $29.0 \pm 27.2$ | $26.7 \pm 25.2$ | $27.6 \pm 26.3$ |

Table 2: **MNIST** Average performances over 5 trials of three FL algorithms: FedAvg, FedRep and FedPer. For each algorithm we compare the vanilla performance with the ones obtained after clustering the clients using the FedOTDD distance, using three different parameters of the spectral clustering algorithm and for a support size of 10 and 100. The number of clients varies from 10 to 100. For this table, datasets from clients **do have** a clear cluster structure

| | | Clustered | | | | | |
| | | Affinity | | Sparse G. (3) | | Sparse G. (5) | |
| | Vanilla | 10 | 100 | 10 | 100 | 10 | 100 |
|---|---|---|---|---|---|---|---|
| FedAvg | | | | | | | |
| 10 | $20.2 \pm 0.6$ | $81.0 \pm 4.2$ | $\mathbf{81.3 \pm 4.5}$ | $78.0 \pm 6.0$ | $77.7 \pm 6.6$ | $71.5 \pm 5.1$ | $72.0 \pm 6.0$ |
| 20 | $25.1 \pm 6.6$ | $71.3 \pm 7.3$ | $\mathbf{72.0 \pm 4.3}$ | $59.5 \pm 3.0$ | $59.5 \pm 5.7$ | $57.0 \pm 4.4$ | $60.5 \pm 2.3$ |
| 40 | $42.5 \pm 10.5$ | $\mathbf{70.8 \pm 13.5}$ | $70.3 \pm 13.3$ | $60.0 \pm 3.7$ | $59.5 \pm 10.6$ | $58.1 \pm 6.3$ | $56.9 \pm 6.1$ |
| 100 | $52.6 \pm 3.9$ | $64.4 \pm 9.6$ | $60.4 \pm 11.3$ | $\mathbf{76.3 \pm 5.4}$ | $68.2 \pm 6.1$ | $67.9 \pm 6.0$ | $65.4 \pm 3.7$ |
| FedRep | | | | | | | |
| 10 | $54.3 \pm 11.2$ | $90.1 \pm 6.7$ | $90.1 \pm 7.5$ | $92.1 \pm 4.2$ | $91.8 \pm 4.6$ | $91.0 \pm 4.4$ | $\mathbf{94.0 \pm 3.1}$ |
| 20 | $75.6 \pm 9.3$ | $\mathbf{87.5 \pm 4.5}$ | $86.1 \pm 2.6$ | $81.4 \pm 8.6$ | $85.1 \pm 6.3$ | $85.3 \pm 7.3$ | $87.1 \pm 5.5$ |
| 40 | $78.0 \pm 6.3$ | $\mathbf{88.0 \pm 4.3}$ | $85.4 \pm 4.8$ | $78.9 \pm 7.9$ | $74.9 \pm 8.7$ | $76.7 \pm 5.6$ | $79.6 \pm 5.7$ |
| 100 | $86.0 \pm 4.8$ | $\mathbf{91.6 \pm 3.1}$ | $90.7 \pm 3.7$ | $89.1 \pm 5.0$ | $84.5 \pm 2.9$ | $86.3 \pm 4.9$ | $84.9 \pm 3.6$ |
| FedPer | | | | | | | |
| 10 | $82.0 \pm 10.1$ | $98.4 \pm 1.4$ | $96.5 \pm 3.5$ | $96.4 \pm 3.5$ | $96.5 \pm 3.6$ | $\mathbf{98.5 \pm 1.4}$ | $98.3 \pm 1.3$ |
| 20 | $90.5 \pm 2.4$ | $92.7 \pm 1.5$ | $95.4 \pm 0.5$ | $93.0 \pm 4.3$ | $\mathbf{96.2 \pm 3.0}$ | $93.8 \pm 2.9$ | $94.5 \pm 2.5$ |
| 40 | $\mathbf{92.3 \pm 1.3}$ | $90.2 \pm 4.7$ | $91.0 \pm 4.9$ | $87.7 \pm 4.1$ | $87.0 \pm 3.7$ | $89.2 \pm 2.3$ | $87.5 \pm 5.4$ |
| 100 | $\mathbf{96.6 \pm 0.9}$ | $\mathbf{96.6 \pm 1.6}$ | $96.4 \pm 2.0$ | $92.1 \pm 3.3$ | $93.0 \pm 2.3$ | $90.2 \pm 4.9$ | $86.9 \pm 1.7$ |
| Average Uplift | - | $18.9 \pm 18.9$ | $18.3 \pm 19.2$ | $15.7 \pm 18.6$ | $14.8 \pm 18.6$ | $14.1 \pm 17.1$ | $14.3 \pm 18.1$ |

Table 3: **MNIST** Average performances over 5 trials of three FL algorithms: FedAvg, FedRep and FedPer. For each algorithm we compare the vanilla performance with the ones obtained after clustering the clients using the FedOTDD distance, using three different parameters of the spectral clustering algorithm and for a support size of 10 and 100. The number of clients varies from 10 to 100. For this table, datasets from clients do not have a clear cluster structure

| | | Clustered | | | | | |
| --- | --- | --- | --- | --- | --- | --- | --- |
| | | Affinity | | Sparse G. (3) | | Sparse G. (5) | |
| | Vanilla | 10 | 100 | 10 | 100 | 10 | 100 |
| FedAvg | | | | | | | |
| 10 | $17.6 \pm 1.1$ | $\mathbf{79.1 \pm 6.3}$ | $78.6 \pm 6.0$ | $61.6 \pm 2.6$ | $69.5 \pm 5.1$ | $72.2 \pm 9.4$ | $72.3 \pm 6.0$ |
| 20 | $22.0 \pm 2.6$ | $\mathbf{75.1 \pm 6.2}$ | $66.9 \pm 9.1$ | $42.6 \pm 4.5$ | $52.4 \pm 17.0$ | $52.2 \pm 8.8$ | $56.2 \pm 13.6$ |
| 40 | $26.1 \pm 7.1$ | $65.9 \pm 7.1$ | $\mathbf{70.1 \pm 5.7}$ | $36.7 \pm 18.3$ | $46.2 \pm 15.7$ | $48.8 \pm 8.3$ | $49.9 \pm 12.1$ |
| 100 | $26.4 \pm 4.3$ | $68.0 \pm 5.1$ | $\mathbf{68.3 \pm 4.7}$ | $37.4 \pm 11.4$ | $44.9 \pm 13.0$ | $39.8 \pm 8.0$ | $43.1 \pm 10.4$ |
| Fedrep | | | | | | | |
| 10 | $82.4 \pm 2.3$ | $\mathbf{91.1 \pm 1.2}$ | $90.7 \pm 1.2$ | $89.4 \pm 0.8$ | $90.3 \pm 1.0$ | $89.7 \pm 2.3$ | $90.0 \pm 1.1$ |
| 20 | $81.8 \pm 1.8$ | $\mathbf{88.1 \pm 2.0}$ | $85.9 \pm 1.4$ | $84.4 \pm 0.5$ | $86.0 \pm 2.1$ | $85.3 \pm 0.5$ | $86.8 \pm 1.4$ |
| 40 | $80.3 \pm 0.8$ | $83.7 \pm 2.0$ | $\mathbf{86.2 \pm 0.9}$ | $81.0 \pm 2.1$ | $82.3 \pm 2.5$ | $81.6 \pm 1.7$ | $82.1 \pm 1.4$ |
| 100 | $75.0 \pm 0.9$ | $\mathbf{79.4 \pm 2.3}$ | $78.5 \pm 1.7$ | $75.2 \pm 2.4$ | $76.3 \pm 1.6$ | $75.4 \pm 1.5$ | $76.9 \pm 1.1$ |
| FedPer | | | | | | | |
| 10 | $82.1 \pm 2.3$ | $\mathbf{93.2 \pm 1.1}$ | $93.0 \pm 0.8$ | $91.7 \pm 0.5$ | $93.0 \pm 0.8$ | $92.3 \pm 2.0$ | $92.7 \pm 1.0$ |
| 20 | $85.4 \pm 2.3$ | $\mathbf{91.0 \pm 1.9}$ | $89.1 \pm 1.8$ | $87.2 \pm 0.5$ | $88.7 \pm 2.5$ | $87.8 \pm 0.9$ | $89.5 \pm 1.9$ |
| 40 | $85.9 \pm 0.8$ | $87.2 \pm 2.2$ | $\mathbf{89.7 \pm 1.4}$ | $82.7 \pm 2.5$ | $85.4 \pm 2.7$ | $84.3 \pm 1.9$ | $84.9 \pm 1.6$ |
| 100 | $82.2 \pm 0.4$ | $\mathbf{85.1 \pm 1.8}$ | $83.4 \pm 2.7$ | $80.3 \pm 2.0$ | $81.3 \pm 1.8$ | $80.9 \pm 1.7$ | $82.5 \pm 1.5$ |
| Average Uplift | - | $20.0 \pm 21.3$ | $19.4 \pm 20.8$ | $8.6 \pm 12.5$ | $12.4 \pm 15.1$ | $11.9 \pm 16.0$ | $13.3 \pm 16.1$ |

Table 4: **CIFAR10** Average performances over 5 trials of three FL algorithms: FedAvg, FedRep and FedPer. For each algorithm we compare the vanilla performance with the ones obtained after clustering the clients using the FedOTDD distance, using three different parameters of the spectral clustering algorithm and for a support size of 10 and 100. The number of clients varies from 10 to 100. For this table, datasets from clients **do have** cluster structure

| | | Clustered | | | | | |
| --- | --- | --- | --- | --- | --- | --- | --- |
| | | Affinity | | Sparse G. (3) | | Sparse G. (5) | |
| | Vanilla | 10 | 100 | 10 | 100 | 10 | 100 |
| FedAvg | | | | | | | |
| 10 | $18.1 \pm 0.7$ | $71.3 \pm 7.3$ | $71.0 \pm 3.4$ | $72.7 \pm 6.2$ | $72.6 \pm 4.1$ | $\mathbf{76.6 \pm 2.6}$ | $72.4 \pm 1.6$ |
| 20 | $23.5 \pm 6.9$ | $\mathbf{71.4 \pm 9.7}$ | $71.2 \pm 7.9$ | $42.5 \pm 4.7$ | $47.8 \pm 4.8$ | $49.7 \pm 4.7$ | $44.4 \pm 8.1$ |
| 40 | $26.6 \pm 5.1$ | $\mathbf{73.4 \pm 15.9}$ | $71.1 \pm 15.0$ | $36.3 \pm 4.5$ | $30.9 \pm 7.1$ | $32.3 \pm 11.6$ | $30.3 \pm 4.6$ |
| 100 | $27.5 \pm 2.0$ | $54.6 \pm 10.1$ | $\mathbf{54.6 \pm 10.2}$ | $27.6 \pm 4.1$ | $29.8 \pm 6.8$ | $29.0 \pm 3.8$ | $28.3 \pm 5.6$ |
| FedRep | | | | | | | |
| 10 | $83.6 \pm 2.2$ | $90.3 \pm 3.1$ | $90.3 \pm 2.4$ | $\mathbf{91.2 \pm 1.6}$ | $91.1 \pm 1.8$ | $91.1 \pm 2.7$ | $\mathbf{91.2 \pm 1.7}$ |
| 20 | $85.3 \pm 2.0$ | $90.7 \pm 2.5$ | $\mathbf{91.5 \pm 2.6}$ | $87.9 \pm 2.0$ | $88.4 \pm 2.2$ | $88.1 \pm 1.4$ | $88.6 \pm 1.8$ |
| 40 | $84.1 \pm 0.8$ | $\mathbf{93.6 \pm 2.9}$ | $93.3 \pm 2.8$ | $84.8 \pm 1.7$ | $84.4 \pm 0.7$ | $84.3 \pm 0.5$ | $85.3 \pm 1.2$ |
| 100 | $77.9 \pm 1.4$ | $91.4 \pm 2.0$ | $\mathbf{91.6 \pm 1.9}$ | $77.8 \pm 1.7$ | $78.0 \pm 2.4$ | $79.0 \pm 1.1$ | $79.4 \pm 1.7$ |
| FedPer | | | | | | | |
| 10 | $83.1 \pm 2.1$ | $92.6 \pm 2.2$ | $92.7 \pm 1.4$ | $93.0 \pm 1.4$ | $\mathbf{93.1 \pm 1.5}$ | $93.0 \pm 2.0$ | $\mathbf{93.1 \pm 1.3}$ |
| 20 | $88.7 \pm 1.7$ | $92.3 \pm 1.8$ | $\mathbf{92.7 \pm 2.4}$ | $89.8 \pm 2.0$ | $90.2 \pm 1.8$ | $90.1 \pm 1.5$ | $90.0 \pm 1.2$ |
| 40 | $88.1 \pm 0.7$ | $\mathbf{94.8 \pm 2.6}$ | $94.6 \pm 2.5$ | $86.0 \pm 2.3$ | $86.5 \pm 0.7$ | $84.9 \pm 3.3$ | $85.7 \pm 1.4$ |
| 100 | $85.1 \pm 0.6$ | $94.0 \pm 1.4$ | $\mathbf{94.1 \pm 1.3}$ | $82.0 \pm 2.4$ | $82.3 \pm 2.2$ | $83.0 \pm 1.1$ | $83.6 \pm 1.6$ |
| Average Uplift | - | $19.9 \pm 18.0$ | $19.7 \pm 17.5$ | $8.3 \pm 15.2$ | $8.6 \pm 15.4$ | $9.1 \pm 16.6$ | $8.4 \pm 15.1$ |

Table 5: **CIFAR10** Average performances over 5 trials of three FL algorithms: FedAvg, FedRep and FedPer. For each algorithm we compare the vanilla performance with the ones obtained after clustering the clients using the FedOTDD distance, using three different parameters of the spectral clustering algorithm and for a support size of 10 and 100. The number of clients varies from 10 to 100. For this table, datasets from clients do not have a clear cluster structure

