# OpenReview forum: "Federated Wasserstein Distance"
_ICLR.cc/2024/Conference — ICLR 2024 poster_

### Official Review · Reviewer_xDsE · 2023-10-18

**Soundness:** 3 good
**Presentation:** 3 good
**Contribution:** 3 good
**Rating:** 8
**Confidence:** 3

**Summary:**

The paper proposes Federated Wasserstein Distance, an algorithm to compute the Wasserstein distance between two measures in the setting of federated learning. Due to the setting, a measure on a client cannot be transferred to the server or the other client to compute the distance. Therefore, the paper utilizes the geometry of Wasserstein distance i.e., triangle inequality and interpolating measure to create transferable measures. To ensure privacy of two interested measures $\mu$, $\nu$, three measures are created $\xi$, $\xi_\mu$, $\xi_\nu$ where $\xi_\mu$ is the interpolating measure between $\xi$ and $\mu$,  $\xi_\nu$ is the interpolating measure between $\xi$ and $\nu$, and $\xi$ is  the interpolating measure between $\xi_\mu$ and $\xi_n$. For a fast computation, McCann’s interpolation is used for constructing the interpolating measure. Experiments on toy examples (Gaussian data) show that FedWD can approximate the Wasserstein distance with a small error. In addition, FedWD is used as the approximation for Wasserstein in coreset classification model, Geometric dataset distances, and  Boosting FL methods. Overall, FedWD shows a "similar" transportation cost as the exact Wasserstein distance.

**Strengths:**

* The paper tackles a new practice setting for computing Wasserstein distance i.e., federated learning. This is the first work that considers computing Wasserstein distance in this setting.
* The usage of the interpolating measures is the key in this setup which is a novel contribution.
* There are some (asymptotically) theoretical guarantees for the approximation from FedWD.
* Experiments are conducted on various applications of Wasserstein distance to show a wide range of applications.

**Weaknesses:**

* Despite interpolating measures being well-defined for general measures, the paper only considers discrete measures setting. In this setting, interpolating measures can be computed in closed form (after knowing the transportation plan). In the continuous setting, this could be not doable (it seems that a transport map or a parametric measure must be used as the proxy in this setting). Considering the continuous setting e.g., with square L2 cost, could make the paper much stronger.
* The gradient approximation scheme should be discussed. For example, If an application wants to estimate the gradient with respect to the support of a measure of the Wasserstein distance, could FedWd yield any approximations for the gradient?

**Questions:**

* Could FedWd be extended to continuous cases?
* Could FedWd yield any approximations for the gradient?
* For some ground costs that are not metric e.g., in domain adaptation application, the corresponding Wasserstein cost is not metric hence there is no triangle inequality. In this case, could we have any solutions?
* How does $t$ affect the performance?

---

> ### Author Response · Authors · 2023-11-15
>
> We thank the reviewer for their positive evaluation of our work and for finding our contribution novel.
>
> > Could FedWd be extended to continuous cases?
>
>
> * In theory, FedWad can be extended to the continuous distributions  since the notion of geodesics is also well-defined in this case  and if a transport map exists then the interpolating measure can be written as $$\mu_t = ( (1-t)Id + tT)_{\\#} \mu_0$$ Hence for a squared L2 cost and continuous distributions, situation for which an optimal map always exists, our approach can still hold. For instance, for general multivariate Gaussians with diagonal covariance matrices, one can compute the barycenter $\mu_t$ of 2 Gaussians in closed form. We provide in the Appendix D.2 an illustration on how the measure $\xi^{(k)}$ evolves in this situation and how the Federated WD converges towards the true WD.
>
> > Could FedWd yield any approximations for the gradient?
>
> * We do not fully understand the question. The reviewer seems to ask if FedWad can be used to approximate the gradient of the Wasserstein distance w.r.t. to the parameters of $\mu$ or $\nu$, and if so, whether the approximation would be accurate enough. If so, then the answer is affirmative and the quality of the approximation depends on the error on the computed transport plan. This is the case for any numerical scheme that computes the Wasserstein distance since any scheme would approximate only the true optimal plan (we use the envelope theorem for computing the gradient).  We note that in our experiments on coresets, the coresets learned by gradient descent on FedWad are visually similar to those obtained with exact WD (Figure 4), and they are useful for yielding a federated Knn that outperforms existing FL algorithms. While these empirical results are encouraging, a theoretical analysis might be needed to rigorously characterize the approximation error on the gradient; this is out of scope of our work.
>
> > For some ground costs that are not metric e.g., in domain adaptation application, the corresponding Wasserstein cost is not metric hence there is no triangle inequality. In this case, could we have any solutions?
>
> * FedWad strongly relies on the fact that the true Wasserstein distance satisfies the triangle inequality. Hence, it is not straightforward to adapt our algorithm to divergences which do not verify the triangle inequality.
>
>
> >How does  $t$ affect the performance?
>
> * the choice of $t$ does **not** affect the performance of the algorithm neither in terms of running time nor in terms of error. At each iteration of the algorithm, we look for *any* element on the geodesic. Hence, any value of $t$ can be chosen. However a deterministic choice of $t$ might facilitate privacy leaks.

---

> ### Comment · Reviewer_xDsE · 2023-11-15
> **Response to authors**
>
> Thank you for your reply,
>
> "Could FedWd yield any approximations for the gradient?"
>
> Yes, I refer to the approximation of the gradient of the Wasserstein distance w.r.t. to the parameters. It is still unclear to me how the gradients of the coreset are computed. I assume that the measure $\varepsilon$ is found then the gradient is taken from $W(\varepsilon$, empirical measure over coreset). Is it correct?

---

> ### Author Response · Authors · 2023-11-15
>
> > It is still unclear to me how the gradients of the coreset are computed. I assume that the measure $\varepsilon$ is found then the gradient is taken from $W(\varepsilon, empirical measure over coreset)$. Is it correct?
>
> Let say, we want to minimize  $$W(f_\theta, \mu)$$ with respect to $\theta$, $f_\theta$ being on server. In a FedWad notation, this mean that we want to solve $$ min_\theta~~W(f_\theta, \xi^K) + W(\mu, \xi^K)$$  Once we have applied FedWad, we have   $W(f_\theta, \xi^K) = \langle P^\star, M_{f,{\xi^K}} \rangle$ where $P^\star$ is the optimal plan and  $M_{f,\xi^K}$ is the matrix of pairwise distances. Based on the envelope theorem the gradient  $\nabla_\theta W$ is $\langle P^\star, \nabla_\theta M_{f,\xi^K} \rangle$ where the gradient is computed componentwise and this gradient is used for updating $f_\theta$ (in pytorch, the gradient is computed by autograding torch.sum(P.detach()*M))

---

> > ### Comment · Reviewer_xDsE · 2023-11-16
> > **Response to the author**
> >
> > Thank you for your reply,
> >
> > It is clear to me about the computation of the gradient now. The FedWD seems to be able to apply to various applications. In addition, the authors have added a discussion on privacy to the revision. Although there are still some limitations as pointed out by other reviewers, I believe the paper is worth being accepted at ICLR. I raised my score to 8, however, I reduced my confidence to 3 since I do not know the specific challenges that the method can suffer in the federated learning setting.
> >
> > Best,

---

### Official Review · Reviewer_71t9 · 2023-10-29

**Soundness:** 3 good
**Presentation:** 3 good
**Contribution:** 3 good
**Rating:** 6
**Confidence:** 4

**Summary:**

The paper motivates the Federated Wasserstein distance that would be useful in situations where we what to know the distance between two distributions while the samples of the two distributions are not shared. The authors propose a converging algorithm that can compute such a Federated Wasserstein distance.

**Strengths:**

I found the problem is well motivated. The theoretical treatment is nice and sound. The numerical experiments are convincing.

**Weaknesses:**

The presentation of the manuscript could be better.

**Questions:**

1. Do we need the two distributions to be discrete and the samples of the two distributions to be the same? Theorem 1 seems to be less general than Theorem 2. If so, the motivation of Theorem 1 should be emphasized and highlighted.

2. I think an experiment where the datasets are unbalanced would be super helpful; say what if a dataset is always a certain times of the other dataset.

3. An experiment that compares the Gaussian & non-Gaussian assumptions to show the convergence speed could be better.

---

> ### Author Response · Authors · 2023-11-15
>
> We thank the reviewer for their positive rating and for finding the problem well motivated, our theoretical development sound and our experiments convincing.
>
> > Do we need the two distributions to be discrete and the samples of the two distributions to be the same? Theorem 1 seems to be less general than Theorem 2. If so, the motivation of Theorem 1 should be emphasized and highlighted.
>
> * In theory, the two distributions do **not** need to be discrete, since the notion of geodesics is well-defined for continuous distributions. However, we need a closed-form expression of the transport map or a closed-form of the Wasserstein barycenter in order to be able to compute any element of a geodesic.
>
>
> * Furthermore, the number of samples in each distribution **can be different**. There is no restriction on this and we have run experiments in which the datasets are imbalanced (one measure as $n$ samples and the other $3n$, see Figure 2)
> * Theorem 1 is indeed less general than Theorem 2. Its goal is to formalize the soundness of the approximate interpolating measure given in Equation 10 compared to the exact formulation in Equation 5. We stated that under certain situations, Equation 10 provides the same result as Equation 5.
>
> > I think an experiment where the datasets are unbalanced would be super helpful; say what if a dataset is always a certain times of the other dataset.
>
> We have run such an experiment on a toy problem and reported the results in Figure 2. We have shown there that the imbalance does not impact much the running time nor the approximation error of FedWad either using the exact interpolating measure or the approximated ones.  In practice, using the approximated interpolating measure leads to better approximation and faster computation.
>
>
>
> > An experiment that compares the Gaussian & non-Gaussian assumptions to show the convergence speed could be better.
>
> * We have included an example of convergence for Gaussians and non-Gaussian distributions. It is reported in the Appendix D.3 and Figure 11. We can note there that for non-Gaussians, the approximation error is reduced by an order of magnitude (vs $1.5$ order of magnitude for Gaussians) for every $2$ early iterations and the rate tends to decrease for the higher iterations.

---

### Official Review · Reviewer_vrdV · 2023-10-30

**Soundness:** 3 good
**Presentation:** 4 excellent
**Contribution:** 3 good
**Rating:** 6
**Confidence:** 4

**Summary:**

The authors propose an algorithm for a federated computation of the wasserstein distance, where the datasets are held by different agents. Intuitively, this algorithm finds a fixed point $(\xi,\xi^\mu,\xi^\nu)$ of the following system of equations:
$$\xi^\mu=BC(\xi,\mu)$$
$$\xi^\nu=BC(\xi,\nu)$$
$$\xi=BC(\xi^\mu,\xi^\nu)$$
where $BC$ is the operation of computing a Wasserstein barycenter. The authors show communication cost, convergence and a theoretical analysis of some special cases. The proposed algorithm is then applied to a variety of problems.

**Strengths:**

I think the authors propose an algorithm which could be valuable in federated learning and other data-science applications in the federated setting. The paper is well written and the concepts and development is easy to follow. I especially found Figure 1 to be very insightful. The Theorems support the claims and are valuable. While the setting of Theorem 3 is arguably a bit restrictive, the result is nonetheless very interesting. Specifically the fact that you can compute the Wasserstein distance in only one communication round!

**Weaknesses:**

Two main desiderata of federated learning are privacy and low communcation cost.
While the problem of communication cost is addressed with (10), the problem of privacy remains largely unanswered.
If the authors convincingly address the problem of privacy leak, I am open to change my recommendation.
If I correctly understand the reasoning at the bottom of page 4, the authors propose to randomly select $t$ such that the server cannot easily infer $d_{\mu,\xi^{(k)}}$. While I agree that this is improves the privacy, I think that the server can, after just two communication rounds, infer $\mu$ and $\nu$. Consider the simplified setting of theorem 3.
If the server knows $(m_{\xi^{(k)}}, m_{\xi^{(k)}\mu})$ and $(m_{\xi^{(k+1)}}, m_{\xi^{(k+1)}\mu})$, she knows that $\mu$ must line on the line through $(m_{\xi^{(k)}}, m_{\xi^{(k)}\mu})$ and on the line through $(m_{\xi^{(k+1)}}, m_{\xi^{(k+1)}\mu})$. These lines must cross at $m_\mu$ by construction. (Here, $m_{\xi^{(k)}\mu}$ denotes the mean of $\xi^{(k)}_\mu$).

**Questions:**

I have two minor comments/questions:
- In Algorithm 1, you use $d_{\mu,\xi^{(k)}}$. While I can infer from the context what it probably means, this symbol has not been used before. I recommend defining it somewhere.
- In the experiments, for the Wasserstein Coreset it does not seem straight forward how your algorithm can be applied to this problem. Could you please explain it to me which agents hold which distributions and where your algorithm enters the computation?

---

> ### Author Response · Authors · 2023-11-15
> **discussing privacy**
>
> We thank the reviewer for finding our work and theoretical developments valuable for the community.
>
> The main criticism of the reviewer is related to privacy issue. While we acknolewdge that having privacy guarantees is an important desiderata in many applications, federated learning does not provide strong guarantee about this: we refer to our answer to all reviewers, as well as our answer to reviewer FgAh, for more explanations.
>
> > I think that the server can, after just two communication rounds, infer $\mu$ and $\nu$.
>
> The attack proposed by the reviewer is based on finding the intersection between the lines joining $\xi^{(k)}$, $\xi_\mu^{(k)}$ and the one joining $\xi^{(k+1)}$, $\xi_\mu^{(k+1)}$.
>
> * In the simplified setting described by the reviewer, we note that our approach needs only 1 communication round while the attack needs 2 rounds. Hence, arguably a simple defense for the client is to allow only 1 query of interpolating measure. In addition, we can also emphasize that the revealed information is an aggregate information on the distribution from which it is difficult to infer  individual samples.
> * In a more general setting (beyond Gaussians), the strategy proposed by the reviewer is supposed to reveal the location of the $\mu$ in the Wasserstein space. However, we believe that the key notion of lines $\xi^{(k)}$, $\xi_\mu^{(k)}$ needs a proper formal definition. Indeed, the definition of the geodesic between $\xi^{(k)}$ and $\xi_\mu^{(k)}$ is known but  it is not clear for us how one should define the extension of this segment beyond $\xi_\mu^{(k)}$ towards $\mu$. Then, assuming that these lines are properly defined, and one can estimate their intersection (in the Wasserstein space) -- which we believe is an hard problem --, what we reveal is a *global* information (the distribution) but not individual samples.
>
>
> * The approximated interpolating measure approach (given in Equation 10 and used in practice) not only facilitates favorable computational complexity and communication costs, but also encourages privacy. Indeed, as stated in Equation 10, the support of the interpolating measure depends  on the data also through a matrix multiplication with the optimal plan. This plan is not revealed to the server, thus making the inference of $\mu$ not straighforward. In addition, when using the approximated interpolating measure, there exists situations in which several distributions $\mu$ lead to the same $\xi_\mu$, increasing the difficulty of inferring the true $\mu$.
>
> > In Algorithm 1, you use $d_{\mu,\xi^{(k)}}$ . While I can infer from the context what it probably means, this symbol has not been used before.
>
> We thank the reviewer for spotting this: $d_{\mu,\xi^{(k)}}$ is the Wasserstein distance between $\mu$ and $\xi_{(k)}$, and we revised our manuscript to clarify this. We use  the same notation as in the text $\mathcal{W}_p(\xi^{(k)},\nu)$.
>
> > In the experiments, for the Wasserstein Coreset it does not seem straight forward how your algorithm can be applied to this problem. Could you please explain it to me which agents hold which distributions and where your algorithm enters the computation?
>
> We added the following discussion in our paper to clarify the setting of this experiment.
>
> * Finding a coreset of $\mu$ aims at finding the support of an empirical distribution that minimizes the problem $$\min_{\{x_i^\prime\}} W(\mu, \frac{1}{n}\sum_n \delta_{x_i^\prime})$$ We propose to solve this problem in a federated learning manner, by computing the Wasserstein distance with FedWad. Hence, in practice, at each iteration $j$ of the coreset optimization algorithm, we update the coreset support based on the distance $d_{\xi^{(K)}, \frac{1}{n}\sum_n \delta_{x_i^j}}$, where $x_i^ĵ$ is the current estimation of the support.
> * For the federated coreset classification model experiment, we have computed the coresets for each empirical distribution  stored on clients. For instance, in Figure 5, we have 1000 clients with about 30 samples per client, and for each client, we learn a Wasserstein coreset based on 2 samples. This leads to 2000 learnt coresets stored on the server.

---

> > ### Comment · Reviewer_vrdV · 2023-11-20
> > **Thank you for your clarifications**
> >
> > I thank the authors for their clarifying responses and for updating the paper.
> > In the edited version, the authors cite the use of a differentially private version of the Wasserstein distance, which should address the issue of privacy leakage. While I would be interested in how DP affects the performance of your algorithm, I acknowledge that this is hard to achieve before the deadline and I can estimate the error introduced by DP from the results by LeTien et al 2019.
> > As DP is a stand way of ensuring privacy in a federated setting, and the proposed method is compatible with it, I will increase my score to 6.

---

### Official Review · Reviewer_FgAh · 2023-11-01

**Soundness:** 2 fair
**Presentation:** 3 good
**Contribution:** 2 fair
**Rating:** 6
**Confidence:** 3

**Summary:**

The paper proposed a novel algorithm for computing the widely used Wasserstein distance (WD) that is in line with federated learning (FL). In particular, as WD is used in a wide range of tasks where privacy is of major concern, FL is a favorable paradigm to conduct its computation in practice. To this end, the paper proposed to utilize the structure of WD and to communicate between server and each client individually. For each iteration of the proposed algorithm, a measure is sent from server to a client, and a barycenter between this and the clients data is computed and sent back. The paper provides theoretical guarantees that the resulting aggregated distance converges to the actual WD in the limit, and in some cases even in 1 step. Computational complexity of the proposed algorithm is also discussed, with certain reduction possible. Various experiments are given to illustrate the performance of the proposed algorithm compared to the vanilla WD, showing that the performance is not compromised.

**Strengths:**

The paper is overall well presented and the writing is clear. The contribution is original to the knowledge of the reviewer.
Some main strengths:
1: The main problem the paper, i.e. FL for WD, is novel and deserves attention. As WD heavily depends on both data sets at the same time, how to minimize the exposure of the raw data is of theoretical importance. The paper proposes a framework that is in line with FL.
2. The paper covers most major aspects of practicality of FL for WD, including both theoretical and computational properties.

**Weaknesses:**

Although the idea of FL for WD is indeed interesting and deserves attention, I have a major concern on whether the proposed algorithm is really compliant with FL principles. Specifically, for each iteration, the server sends a distribution $\xi$ to client with data $\mu$, then a $t$-barycenter $\xi_\mu$ is sent back to the server. By structure of WD geodesic, with knowledge of $\xi$ and $\xi_\mu$ it is already very immediate to reconstruct $\mu$. For instance, if $\xi = \frac{1}{n}\sum_i \delta_{x_i}$ and $\mu = \frac{1}{n}\sum_i \delta_{y_i}$ are both $n$ points uniformly distributed, with optimal correspondence being $T:x_i\to y_i$, then $\xi_\mu = \frac{1}{n}\sum_i \delta_{(1-t)x_i+ty_i}$, and the optimal correspondence between $\xi$ and $\xi_\mu$ will be $T':y_i\to (1-t)x_i+ty_i$. Thus by computing $T'$ between $\xi$ and $\xi_\mu$ on server, the server gains full information of the raw data. Surely $t$ is needed, but can also be inferred from the ratios of WDs between $\xi$ and $\mu$, and between $\xi$ and $\xi_\mu$ (to be fair the paper claimed that WD is not necessary to report to the server). The reviewer does not claim full expertise in FL, but does believe that this not fully in line with the FL paradigm, which seeks to minimize the revealing of data from the client to the server. It would be great if this point can be addressed by the authors.

**Questions:**

The main question is: under an FL setting, is it in line to essentially reveal raw data to the server? As I remarked above, the server essentially have all of the raw data once it receive both responses, so it is unclear why it cannot just compute WD thereafter. Please see above (section Weaknesses) for more details.

Certainly some amount of information has to be revealed to the server in FL, as client is communicating with the server with a locally trained object. Thus one possible way to address this is to perhaps quantify the privacy guarantee in the framework. I'm happy to raise score once this is fully addressed.

---

> ### Author Response · Authors · 2023-11-15
> **discussing the privacy aspect.**
>
> We thank the reviewer for acknowledging the importance of the problem and the novelty of our contribution. We are grateful for their very insightful questions.
>
> > Under an FL setting, is it in line to essentially reveal raw data to the server?
>
> While we do not have formal guarantees on the robustness of FedWad to privacy leaks, FedWad is in line with the perspective on Federated Learning, as described in Kairouz et al., 2019: *"federated learning (...) provides a level of privacy to participating users through data minimization: the raw user data never leaves the device, and only updates to models (e.g., gradient updates) are sent to the central server. (...) there is still no formal guarantee of privacy in this baseline federated learning model. For instance, it is possible to construct scenarios in which information about the raw data is leaked"*.
>
> As we argue below, we believe it is not straightforward to infer the distribution $\mu$ from the sole information that navigates through the servers, especially when using approximated interpolating measures with random values of $t$.
>
> > The server sends a distribution $\xi$ to client with data $\mu$, then a $t$-barycenter $\xi_\mu$ is sent back to the server. By structure of WD geodesic, with knowledge of $\xi$  and $\xi_\mu$ it is already very immediate to reconstruct $\mu$.
>
> The main question and weakness raised by the reviewer is based on the assumption that there exists a simple attack that reveals the raw data. However, we propose below a rebuttal to this attack and would be happy to further discuss this aspect with the reviewer.
>
> * The attack proposed by the reviewer assumes the knowledge of some important information: $t$, the Wasserstein distances and the number of samples in $\mu$. As acknowledged by the reviewer themselves, none of these elements are supposed to be revealed to the server during the iterations.
> * Another insight that we learn from the reviewer's question is that our approximated interpolating measure approach (given in Equation 10) not only helps improve computational complexity and communication costs, but also facilitate robustness to privacy leaks. Indeed, as stated in Equation 10, the support of the interpolating measure depends on the data through a matrix multiplication with the optimal plan. This plan is not revealed to the server, thus making the inference of $\mu$ not straighforward. In addition, when using the approximated interpolating measure, there exists situations in which several distributions $\mu$ lead to the same $\xi_\mu$, increasing the difficulty of inferring the true $\mu$.
>
> Hence, while we do **not** have a formal result guaranteeing that FedWad does not reveal any information, we do not think that the situation described by the reviewer can lead to disclosure of the raw data. Furthermore, in practice, we use approximated interpolating measure which employs a important piece of information -- the transport plan -- that is not revealed by the client.
>
> If a formal guarantee of privacy is needed, then we can resort to differential privacy and consider a differentially private Wasserstein distance instead.
>
>
> [1] Kairouz, P., McMahan, H. B., Avent, B., Bellet, A., Bennis, M., Bhagoji, A. N., ... & Zhao, S. (2021). Advances and open problems in federated learning. Foundations and Trends® in Machine Learning, 14(1–2), 1-210.

---

> > ### Comment · Reviewer_FgAh · 2023-11-20
> > **Thanks for your response**
> >
> > I thank the authors for the response, and acknowledge that there are key elements missing in order to fully infer the raw data.
> >
> > However I have one last concern from the rebuttal: the paper claims that the number of samples n is not revealed, and some fixed small number $S$ is preferred as the support size of the iterations $\xi$. While this seems to add more privacy and alleviates complexity, it's unclear why it makes theoretical sense (e.g. consistency), as using S different from n can be substantially different from solving (9), thus potentially hindering the guarantees such as convergence. Is this an issue?
> >
> > When checking the convergence result Theorem 2, the proof is also very confusing: the reviewer personally thinks that the convergence is expect even if $t$ is always randomly chosen, but the proof does not seem to give such study. Rather, the sentence "Hence, by choosing t = 0 and t = 1 in the definition of the interpolating measure" is really confusing, as 1) explicit choice of such t is not allowed as remarked in the proof, and 2) there does not seem to be a mechanism to drive $t$ to 0 and 1. Thus the proof does not seem right at the presented form.
> >
> > Once the above is addressed, I'm happy to raise the score.

---

> > > ### Author Response · Authors · 2023-11-21
> > >
> > > Thanks for your comments and remarks.
> > >
> > > > and some fixed small number $S$ is preferred as the support size of the iterations $\xi$. While this seems to add more privacy and alleviates complexity, it's unclear why it makes theoretical sense (e.g. consistency), as using S different from n can be substantially different from solving (9), thus potentially hindering the guarantees such as convergence. Is this an issue?
> > >
> > > When using the exact interpolating measure approach, we initialize the algorithm with a $\xi^{(0)}$ of any size, then the size of $\xi^{(k)}$ grows depending on the support of the optimal plan. For instance, if $\xi^{(0)}$ is of size $S$, and $\mu$ and $\nu$ of size $n$, the support of $\xi^{(1)}$ can be at most of size 2n + 2S + 3.
> > >
> > > When we use the approximated interpolating measure, we indeed gain in time complexity and in privacy at the expense of an approximation error. We agree that our proof of convergence holds only for the exact interpolating measure approach. Nonetheless in practice, we show that this approximated approach is more robust (see Figure 2) and having larger support of the approximation does not necessarily lead to better perfomances. We have reported in the Appendix Table 2 to 5 detailed results comparing the benefit of using S=10 vs S=100, for computing similarity of datasets accross clients. Our results show that using $S=10$ is  sufficient for having a strong boost in performance of the tested FL algorithms.
> > >
> > > > When checking the convergence result Theorem 2, the proof is also very confusing: the reviewer personally thinks that the convergence is expect even if  is always randomly chosen, but the proof does not seem to give such study.
> > >
> > > Yes, indeed the convergence occurs even when $t$ is randomly chosen since any element of the geodesics is going to minimize Equation 9.
> > >
> > > The part mentioned by the reviewer is indeed confusing. What we meant is that because of all possible choices of $t$, there exists an infinite number of triplets $(\xi_\mu^{(\infty)}, \xi_\nu^{(\infty)}, \xi^{(\infty)})$ than can lead to the stationary point $A^{(\infty)}$. And, **at convergence**,  $(\xi_\mu^{(\infty)}, \xi_\nu^{(\infty)}, \xi^{(\infty)})$ are fixed points of the algorithm and we show that $\xi^{(\infty)}$ is an interpolating measure of $\mu$ and $\nu$ in addition to be an interpolating measure of $\xi_\mu^{(\infty)}$ and $\xi_\nu^{(\infty)}$. Indeed,  for any $\xi^{(\infty)}$, $\xi_\mu^{(\infty)}$ can be  any interpolating measure between $\mu$ and $\xi^{(\infty)}$. The same reasoning holds for $\xi_\nu^{(\infty)}$ and $\nu$. Then since $\xi^{(\infty)}$ is an interpolating measure of $\xi_\mu^{(\infty)}$ and $\xi_\nu^{(\infty)}$ and $\mu$ and $\nu$ are possible choices of those interpolating measures, it yields that $\xi^{(\infty)}$ is indeed an interpolating measure of $\mu$ and $\nu$.
> > >
> > > Proof has been corrected accordingly.

---

> > > > ### Comment · Reviewer_FgAh · 2023-11-21
> > > > **Thanks for the clarification**
> > > >
> > > > With the above clarification, I acknowledge that the questions are all addressed. I thus raise my score to 6.

---

### Author Response · Authors · 2023-11-15
**General comments**

**General Comments** Thanks to the reviewers for their questions and insights. Given their comments we made  few updates to the paper which have been mostly reported to the appendix as additional experiments.

**Preamble to rebuttal.** The main idea of Federated Learning (FL) is to learn a model $f$ from a number of datasets under the constraint that each of these training datasets is secluded in its own device/client. The concept of privacy carried by FL is for the datasets (or any part of them) to never leave their device during training. The methodology we propose, FedWad, obeys this FL principle: FedWad computes the Wasserstein distance between two distributions $\mu$ and $\nu$ stored on two different devices, without requiring them to leave their devices.

While reviewers FgAh and vrdV share concerns about the lack of privacy guarantees of our methodology, therefore questioning the FL nature of our contribution, we would like to point out that *FL algorithms are in essence not guaranteed to be immune to all sorts of privacy attacks* (see Kairouz et al, 2019, Section 4). This aspect justifies existing studies and methodologies aiming at making FL more robust to privacy leaks, and confirms that FedWad computes the Wasserstein distance in a FL manner.

That being said, and as we explain in our answers to reviewers, FedWad goes a bit further than vanilla FL in terms of privacy: the use of random values of $t$ and approximate interpolating measures brings additional obfuscation to $\mu$ and $\nu$. A theoretical analysis to characterize when FedWad preserves privacy is an interesting yet challenging problem, which we leave for future work.


[1] Kairouz, P., McMahan, H. B., Avent, B., Bellet, A., Bennis, M., Bhagoji, A. N., ... & Zhao, S. (2021). Advances and open problems in federated learning. Foundations and Trends® in Machine Learning, 14(1–2), 1-210.

---

> ### Author Response · Authors · 2023-11-20
>
> We would like to thank all of the reviewers for their time and thoughtful efforts in providing detailed and constructive feedbacks.  We have carefully considered their comments and have prepared a rebuttal addressing each one. We are open to further discussions and are more than willing to respond to any additional comments or queries they may have.
>
> Thanks to  reviewer **xDsE** for our discussions and for acknowledging the relevance of our work.
>
> We would like to take this opportunity to provide another connection with some existing federated learning algorithms and our approaches in term of data leak. If we assume a FL that looks at solving a regression problem with quadratic loss in which clients share local gradients then, clients send to server
> * $X^\top(y - Xw)$,  where $X$ and $y$ are local data
>
> In our case, clients send to server the support of the  approximated interpolating measures which is
> * $(1-t)X + t n PY$, where $X$ and $Y$ are the support of $\xi$ and $\mu$, and  $P$ the transport plan (which is not revealed)
>
>
> Hence, while we do not have a formal proof of data privacy, we can assume that our approach is at least as compliant with federated learning as some existing FL algorithms.

---

### Meta-Review · Area_Chair_6fzv · 2023-12-06

**Metareview:**

The paper studies the federated learning of Wasserstein distance, designed for scenarios where it's necessary to measure the distance between two distributions without sharing their respective samples. The authors have developed an algorithm that can compute the Wasserstein distance by communication between nodes and proved its convergence.

A key innovation in this research is the use of interpolating measures, a unique contribution to the field. To demonstrate the versatility and broad applicability of the Wasserstein distance, the authors conduct experiments across various applications, showcasing its wide-ranging utility.

Some reviewers have concern on the privacy issue in the proposed algorithm as it may cause data information leaked to other nodes.

**Justification For Why Not Higher Score:**

Some reviewers have concern on the privacy issue in the proposed algorithm as it may cause data information leaked to other nodes.

**Justification For Why Not Lower Score:**

This paper is novel enough. In fact, a key innovation in this research is the use of interpolating measures, a unique contribution to the field.

---

### Decision · Program_Chairs · 2024-01-16

Accept (poster)